# Hematite: A Good Catalyst for the Thermal Decomposition of Energetic Materials and the Application in Nano-Thermite

**DOI:** 10.3390/molecules28052035

**Published:** 2023-02-21

**Authors:** Yu Li, Jia Dang, Yuqiang Ma, Haixia Ma

**Affiliations:** Xi’an Key Laboratory of Special Energy Materials/School of Chemical Engineering, Northwest University, Xi’an 710069, China

**Keywords:** Fe_2_O_3_, thermal decomposition, energetic materials, aluminothermic reaction

## Abstract

Metal oxides (MOs) are of great importance in catalysts, sensor, capacitor and water treatment. Nano-sized MOs have attracted much more attention because of the unique properties, such as surface effect, small size effect and quantum size effect, etc. Hematite, an especially important additive as combustion catalysts, can greatly speed up the thermal decomposition process of energetic materials (EMs) and enhance the combustion performance of propellants. This review concludes the catalytic effect of hematite with different morphology on some EMs such as ammonium perchlorate (AP), cyclotrimethylenetrinitramine (RDX), cyclotetramethylenete-tranitramine (HMX), etc. The method for enhancing the catalytic effect on EMs using hematite-based materials such as perovskite and spinel ferrite materials, making composites with different carbon materials and assembling super-thermite is concluded and their catalytic effects on EMs is also discussed. Therefore, the provided information is helpful for the design, preparation and application of catalysts for EMs.

## 1. Introduction

Iron oxide, with the formula Fe_2_O_3_, has characteristics of abundant natural reserves, non-toxicity, strong corrosion resistance, low preparation cost and excellent visible light absorption ability [1,2]. It usually has four common crystal forms: *α*-Fe_2_O_3_ (corundum structure, R3c space group), *β*-Fe_2_O_3_ (twin structure, Ia3 space group), *γ*-Fe_2_O_3_ (spinel structure, Fd3m space group) and *ε*-Fe_2_O_3_ (pna 21 space group) [3], which is widely used in optoelectronics [4], gas sensors [5], lithium ion cells and batteries [6], water treatment [7], and solid propellants [8]. Due to the crystal transformation, the structure is different; thus, the application of iron oxide with different crystal form is quite different. Among these crystal forms, hematite (*α*-Fe_2_O_3_) is widely used in many fields, such as in Li-ion batteries [6], photoelectrochemical water splitting [9] and as combustion catalysts for solid propellants [10], etc., because of its favorable band position (approximately 2.1 eV), inertness, low toxicity, low cost, and natural abundance.

As many MOs, hematite with a nano size has many different characteristics from the block-sized materials. Researchers carried out many works on the preparation of the nano hematite and obtained many fantastic morphologies including nano-rods, nano-grids, nano-ribbons [11], nano-wires [12], nano-tubes [13], nano-rings [14,15], nano-fibrous [16], hollow spheres [17], etc. Yang et al. [18] directly heated and oxidized iron substrates into ordered *α*-Fe_2_O_3_ nano-ribbons and nano-wire arrays in an oxygen atmosphere. Tang and collaborators [19] successfully fabricated porous *α*-Fe_2_O_3_ flower-like nanostructures. Gotić et al. [20] successfully achieved the morphological transformation of *α*-Fe_2_O_3_ from spindles, pseudospheres, nano-tubes to nano-rings by adding divalent metal ions Mn^2+^, Cu^2+^, Zn^2+^, and Ni^2+^ using hydrothermal method. It is found that the introduction of divalent metal cations changes the type of hydroxyl groups on the (001) plane, which makes *α*-Fe_2_O_3_ preferentially grow along the c-axis. Lv et al. [21] systematically studied the effect of F^−^ and SCN^−^ on the growth of *α*-Fe_2_O_3_ particles and successfully prepared *α*-Fe_2_O_3_ hexagonal nano-rings. The adsorption properties of F^−^ on the (100) plane largely slow down its growth along the (100) direction. SCN^−^ can induce directional dissolution of *α*-Fe_2_O_3_ nanocrystals along the [−221] direction [21,22]. Zhang et al. [23] prepared *α*-Fe_2_O_3_ nanocrystals with various morphologies by adjusting the pH value of the K_3_[Fe(CN)_6_] solution at 140 °C, including snowflake, pair-microplate, dumbbell, cylindrical and spindle three-dimensional structures. Generally speaking, the nano hematite, with different morphology, size and shape, can be achieved by controlling the reaction time, temperature, pH, introducing ligands (such as surfactants, inorganic anions and cations), changing the solvent types (such as common organic solvents, ionic liquids) and other conditions through changing the growth rate of different crystal planes. For further requirement of enhancing application, people also try to make some promotion of hematite, including combining hematite with some carbon materials, making perovskite ferrite, spinel ferrite and super thermites, etc. This paper provides the preparation and application of the hematite-based materials and their catalytic effect for EMs. The effects of the preparation of hematite and ferrite with different morphologies on the thermal decomposition of ammonium perchlorate (AP), cyclotrimethylenetrinitramine (RDX), cyclotetramethylenete-tranitramine (HMX) and other energetic materials (EMs) was studied using thermal analysis technique, and different carbon materials used for preparing composite materials to enhance the catalytic properties of hematite-related materials on EMs were stated. For future application, it is necessary to explore new hematite composited and develop new ideas suitable for industrialization requirements.

## 2. Preparation of Hematite with Different Morphology and Their Catalytic Effect on EMs

The preparation methods of pure *α*-Fe_2_O_3_ mainly include precipitation method, gas phase method, solid phase method, sol-gel method, hydrothermal method, hydrolysis solution, solvothermal method, etc. Usually, the morphologies of hematite prepared by different methods are quite different. The differences in the microscopic structural units manifest anisotropy on the macroscopic scale through thousands of superpositions, and the crystals exhibit different physical and chemical properties in different directions [24,25,26]. The atomic arrangement and coordination on the surface of nanoparticles determine the properties of nanomaterials, so properties of nanocrystals can be regulated by controlling the morphology [27,28,29,30]. Therefore, the methods for synthesizing nano-microcrystals with various geometries and different exposed faces raise much research interest [31,32,33]. This section covers the synthesis of hematite with different morphologies and their catalytic effects on EMs.

### 2.1. Preparation of Polyhedral Hematite and the Effect on EMs

The polyhedral structure of *α*-Fe_2_O_3_, the most stable iron oxide under environmental conditions, has attracted much attention due to its various potential applications. Lv et al. [34] synthesized well-dispersed single-crystal dodecahedral and octahedral *α*-Fe_2_O_3_ particles by a facile hydrothermal method assisted by F anion (the morphology of dodecahedral *α*-Fe_2_O_3_ is shown in Figure 1a,b). The dodecahedral particles are hexagonal bipyramid in shape and surrounded by twelve equivalent (101) planes. The only difference between the preparation conditions of dodecahedral and octahedral particles is the concentration of F^−^. Low F^−^ concentrations are favorable for the growth of octahedral particles, high F^−^ concentration contributes greatly to the growth of dodecahedral particles, while without F^−^, the product is quasi-spherical (particle size about 100 nm). If the F^−^ concentration exceeds 26.5 mM, the particle size became non-uniform. With a higher F^−^ concentration of 28 mM, FeOOH nanorods appeared.

Lv et al. [35] later synthesized dodecahedral *α*-Fe_2_O_3_ particles surrounded by twelve (101) faces with the assistance of F^−^ and controlled the particle size to be around 150 nm by affecting the nucleation process of *α*-Fe_2_O_3_.The thermal behavior using a differential scanning calorimeter (DSC) technique with pure AP and a mixture of AP with 2% *α*-Fe_2_O_3_ particles was tested. In the presence of *α*-Fe_2_O_3_ dodecahedral nanoparticles, the LTD (low temperature decomposition peak) at 315 °C almost disappeared, and the HTD (high temperature decomposition peak) shifted to a lower temperature, indicating that the *α*-Fe_2_O_3_ dodecahedral nanoparticles can promote the thermal decomposition of AP. The ordinary *α*-Fe_2_O_3_ nanoparticles with an average diameter of 100 nm with a similar specific surface area to the synthesized *α*-Fe_2_O_3_ dodecahedral nanoparticles was also used to test the thermal decomposition of AP. Comparing curves of b and c, the exposure of the dodecahedral (101) facets was supposed to enhance the catalytic properties of the particles. The curves of b and d indicate that the *α*-Fe_2_O_3_ of the same morphology with nano-particles promoted the exothermic heat of AP more than the micro particles. Thus, the exposed facets were supposed to have much influence on the exothermic decomposition of AP.

Xu et al. [36] found that the micro-octahedra hematite particles can be obtained using FeCl_3_·6H_2_O as the precursor, dissolving in aqueous formamide solution by a hydrothermal in a 22 mL Teflon-sealed autoclave and stored at 160 °C for 24 h. While the hematite of monodisperse nanorods can be obtained following the same procedure if the reaction time shortened to 12 h.

The catalytic performance of the obtained different *α*-Fe_2_O_3_ on the thermal decomposition of AP was studied by DSC. The DSC curves of AP decomposition with micro-octahedrons, irregular particles and nanorods with different *α*-Fe_2_O_3_ samples indicated that all the HTD (436.4 °C) shifted to lower temperatures under the hematite of different morphologies. The prepared *α*-Fe_2_O_3_ nanorods exhibited better catalytic activity than the other two samples, and the catalytic performance of irregular particles was slightly better than that of micro-octahedral particles. The authors suggested that the smaller-sized particles may have better performance. Since the three morphologies and sizes in the references are different, we cannot conclude from the above conclusions that the morphology of the nanorods must have a good catalytic effect.

Our group [37] also synthesized well-shaped single-crystal *α*-Fe_2_O_3_ particles by a hydrothermal method. The mixture of K_4_Fe(CN)_6_·3H_2_O, sodium carboxymethyl cellulose solution (CMC-Na), poly(vinylpyrrolidone) (PVP), and N_2_H_4_·H_2_O solution were sealed in a 40 mL Teflon-lined autoclave and heated at 200 °C for 6 h to obtain the tetrakaidekahedral hematite.

The structure characterization is shown in Figure 2. The synthesized Fe_2_O_3_ nanoparticles are uniform in size, and the particle size is 200–250 nm wide and 300–350 nm long. The concentration of K_4_Fe(CN)_6_·3H_2_O was found to have important effect on the formation of morphology. When the used K_4_Fe(CN)_6_·3H_2_O is 0.2 mmol, the oblique parallelepiped iron oxide was obtained.

Further study [38] about the catalytic effect of the tetrakaidekahedral hematite compared to the grainy hematite on the thermal decomposition of hexanitrohexaazaisowurtzitane (HNIW, CL-20) indicates that the exothermic peak temperatures of CL-20 with tetrakaidecahedral-Fe_2_O_3_ and grainy-Fe_2_O_3_ decreased by 5.53 and 4.95 K, respectively (Figure 3). Further non-isothermal thermal decomposition kinetic studying indicated that the tetrakaidekahedral nano-Fe_2_O_3_ did not change the thermal decomposition mechanism of CL-20, while the grainy nano-Fe_2_O_3_ changed the thermal decomposition mechanism of CL-20. From the decrease in the thermal decomposition peak temperature and the corresponding kinetic evaluation, the tetrakaidekahedral nano-Fe_2_O_3_ exhibits better catalytic activity on CL-20 than the grainy nano-Fe_2_O_3_.

### 2.2. Preparation of Granular Hematite and the Effect on EMs

To study the effect of different iron precursors on the particle size of hematite and its effect on nitrocellulose (NC), Benhammada et al. [39]. successfully prepared *α*-Fe_2_O_3_ nanoparticles by a simple and direct hydrothermal method using three different precursors (FeCl_3_, Fe(NO_3_)_3_ and Fe(SO_4_)_2_). In brief, a stoichiometric amount of iron precursor was dissolved in distilled water under stirring and the solution of ammonia (34%) was used to adjust pH = 8, then in an autoclave and heated at 180 °C for 24 h. The spherical nanoparticles with 110 ± 9, 90 ± 6 and 80 ± 7 nm corresponding to FeCl_3_, Fe(NO_3_)_3_ and Fe(SO_4_)_2_, respectively, were obtained. The catalytic activity of the as-prepared nanoparticles for the thermal decomposition of NC was investigated by DSC. All curves show only one exothermic peak, corresponding to the breaking of the O-NO_2_ bond.

The as-synthesized nanocatalysts with different sizes have a slight effect on the peak decomposition temperature of NC, while the activation energy is apparently reduced. The Fe_2_O_3_ prepared using ferric chloride as a precursor has the greatest catalytic effect on NC.

Our group [39,40] successfully prepared granular nanoscale *α*-Fe_2_O_3_ particles by a simple hydrothermal method using FeCl_3_·6H_2_O, urea, glycine in distilled water at 160 °C for 10 h. The SEM and TEM images of the prepared granular *α*-Fe_2_O_3_ nanoparticles with a small average particle size of 210 nm are shown in Figure 4a,b.

The catalytic effect of granular nanoscale hematite was studied by DSC technique. The obtained DSC curves are shown in Figure 4c. The thermal decomposition peak of *α*-Fe_2_O_3_/NC composites was almost equal with NC, but the activation energy (*E*_a_) of the composite decreased by 15.37 kJ·mol^−1^. Further detailed thermal decomposition mechanism about the composites and NC suggested that the O-NO_2_ bond cleavage, the condensed-phase decomposition and even the reaction of NO_2_ and HCHO gases were promoted by the as-prepared *α*-Fe_2_O_3_. The hematite nanoparticles were supposed to promote the O-NO_2_ bond cleavage, the condensed-phase decomposition and the reaction of NO_2_ and HCHO gases, thus, accelerate the thermal decomposition reaction rate of NC.

Elbasuney et al. [41] prepared high crystalline, monodisperse Fe_2_O_3_ nanoparticles with an average particle size of 3.39 nm using a hydrothermal method, then the obtained Fe_2_O_3_ was integrated into HMX by a co-precipitation technique. The effect of Fe_2_O_3_ (1 wt%) on the pyrolysis of HMX was investigated using DSC. The main exothermic decomposition peak of HMX decreased from 285 °C to 272 °C and the decomposition enthalpy increased from 1016 J·g^−1^ to 1750 J·g^−1^. Mechanistic analysis revealed that Fe_2_O_3_ can convert HMX pyrolysis from C-N cleavage to hydrogen atom abstraction by releasing active surface OH radicals. Furthermore, the released NO_2_ and CH_2_O can be adsorbed on the nanocatalyst surface, providing high decomposition enthalpy.

### 2.3. Preparation of Rod-Shaped Hematite and the Effect on EMs

The heat treatment temperature always affects the phase transition process of amorphous *δ*-FeOOH. Zhang et al. [42] prepared *α*-FeOOH nanorods by hydrothermal reaction at 100 °C for 6 h using Fe(NO)_3_·9H_2_O and KOH as raw materials. Different porous single crystal *α*-Fe_2_O_3_ nanorods were obtained after the samples were heat-treated at 300, 350, 450 and 600 °C. The catalytic effect of the prepared nano-rod samples on AP was investigated by differential temperature analysis (DTA). The *α*-Fe_2_O_3_ nanorods obtained by heat treatment at 350 °C was proved to reduce the pyrolysis temperature of AP by 71.4 °C.

To gain insight into the intrinsic function of catalysts in AP decomposition and guide the design of catalysts. Zhou et al. [43] proposed a new method to quantitatively evaluate the catalytic ability of MOs for AP. Two metrics, intrinsic peak temperature (T^*^_HTD_) and catalytic density (ρ_T_^*^), are introduced during pyrolysis to decouple the activity and the number of catalyst sites. Hematite nanostructures with different shapes were used as model catalysts to verify the feasibility of this method. The different morphology of Fe_2_O_3_ was obtained using different precursors and different preparation methods. Rod-like Fe_2_O_3_ NPs were obtained using Fe(NO_3_)_3_ and NaOH as the precursors, after being sealed in the Teflon reactor at 180 °C for 40 min and calcined at a tube furnace at 350 °C for 4 h. The rhombic Fe_2_O_3_ NPs were obtained using FeCl_3_ and urea as precursors, after the solution was refluxing at 95 °C for 4 h and calcined at a tube furnace at 500 °C for 3 h. The pseudo-cubic Fe_2_O_3_ NPs were obtained using Fe(NO_3_)_3_·9H_2_O and PVP by hydrothermal method at 180 °C for 3 h and annealing at 600 °C for 90 s. The AP decomposition HTD by DSC of 2 wt% ro-Fe_2_O_3_, rh-Fe_2_O_3_ and pc-Fe_2_O_3_ were 358.1, 381.5 and 395.0 °C, respectively, which were 76.8, 53.4 and 39.9 °C lower than that of pure AP decomposition (434.9 °C). The rod-like *α*-Fe_2_O_3_ exhibited the best catalytic performance for the thermal decomposition of AP, with the lowest T^*^_HTD_ of 345.1 °C and the highest ρ_T_* of 0.544, which implies the highest number and activity of catalytic sites.

### 2.4. Preparation of Hematite with Other Morphologies and the Effect on EMs

As an important environmentally friendly MOs, in addition to the common morphology, hematite nanostructures with various shapes have also attracted extensive attention to meet better requirements.

Nano-disc morphology: Shi et al. [44] successfully synthesized hematite hexagonal nano-discs with dominant (001) and lateral (110) planes by introducing ethylene glycol (EG) into a hydrothermal system. The results show that the proper addition of EG in the hydrothermal system creates equilibrium conditions for the nucleation and growth of crystals, resulting in the formation of uniform *α*-Fe_2_O_3_ hexagonal nano-discs with major (001) crystal planes.

The DTA curves of AP and AP-based composites samples indicated that the *α*-Fe_2_O_3_ hexagonal nano-discs exhibited superior catalytic performance compared with the *α*-Fe_2_O_3_ irregular particles. Under the action of *α*-Fe_2_O_3_ hexagonal nano-discs, the exothermic peak induced by the pyrolysis of AP [36] were shifted from 516.6 °C to lower temperatures of 356.0 °C and 394.0 °C. While *α*-Fe_2_O_3_ irregular particles only induce a decrease from 516.6 °C to 419.8 °C. The two shifts to lower peak temperature can be attributed to the different catalytical active lattice and composition of the exposed (001) and (110) crystalline planes due to their different planes.

Hexagonal cone (HC) morphology: Sharma et al. [45] developed a facile and environmentally friendly method for the synthesis of *α*-Fe_2_O_3_ in the HC form by adding neem leaf extract to an aqueous solution of ferric chloride. The morphology and microstructure of the synthesized *α*-Fe_2_O_3_ HCs were investigated by Field Emission SEM (FESEM) measurement, as shown in Figure 5. The nanostructures are hexagonal pyramid shaped with an average diameter of 400–500 nm. Then, the synthesized *α*-Fe_2_O_3_ HCs were used as burn rate enhancers for thermal decomposition of AP and combustion of composite solid propellant. From the thermal analysis (Figure 6), the addition of the synthesized *α*-Fe_2_O_3_ HCs significantly decreased the decomposition temperature of AP by 75 °C, indicating that the green synthesized *α*-Fe_2_O_3_ HCs have an effective catalytic effect on the thermal decomposition of AP. The *α*-Fe_2_O_3_ HCs can simultaneously accelerate LTD and HTD. Further kinetic studies show that the *E*a^*^ value for the ignition is much lower, which is due to the efficient adsorption of gaseous products on the green-synthesized *α*-Fe_2_O_3_ HCs during the dissociation of AP, and thus *α*-Fe_2_O_3_ HCs significantly promotes the gaseous reaction.

For reactive precipitations, they are usually fast or transient reaction processes, with nucleation induction periods typically around 1 ms. The rate of nucleation and crystal growth depends largely on the supersaturation of the reaction environment. Cao et al. [46] heated FeCl_3_·6H_2_O and urea aqueous solution at 95 °C in two different tanks. Then, the two solutions were sprayed into the packing of a rotating packed bed (RPB) to get red precipitate. After transferred into a thermostat bath, the reaction was kept going at 95 °C for 4 h under stirring and refluxing. Finally, the red product was obtained after the red precipitate cake was filtered, washed and calcined in ambient atmosphere at 500 °C for 3 h and drying. When the high gravity level (G, the ratio of centrifugal acceleration generated by the rotating packing to the local gravity acceleration) of the RPB reactor equals 68, the uniform rhombohedral sphere can be formed. The DSC results indicated that the addition of 2 wt% of 84 nm *α*-Fe_2_O_3_ can decrease the LTD and HTD temperatures of AP by 14.4 °C and 53.4 °C, respectively, and increase the heat released from AP from 864 J·g^−1^ to 1235 J·g^−1^.

Table 1 summarizes the preparation methods, thermal decomposition temperature (*T*_p_), reaction heat (Δ*H*), and apparent activation energy (*E*) of these hematite with energetic composites of different morphologies reported in recent years. The morphologies and particle sizes have a great influence on the thermal decomposition process of EMs. from the decrease in the decomposition temperature and activation energy.

## 3. The Preparation of Ferrite and the Application on the EMs

### 3.1. Perovskite Ferrite

Perovskite type of metal oxides, with the general formula of ABO_3_ [47], are compounds composed of two or more simple oxides with high melting points. The structure is shown in Figure 7 (“A” is an alkaline-earth metal, “B” is a transition metal, red sphere represent oxygen atoms), A site is usually an alkaline earth metal or rare earth metal cations, and B position is always transition metal cations. It can be partially substituted by other metal cations with similar radii to keep its crystal structure basically unchanged [48]. Tanaka et al. [49] proposed the main strategies of designing this kind of materials as follows: (1) select the B site element that mainly determines the catalytic activity; (2) control the valence and vacancy by selecting the A site element, including partial substitution; (3) The synergistic effect of the constituent elements is mainly the B site transition elements; (4) enlarge the specific surface area by forming small particles or dispersing on the carrier; (5) add precious metals and their proper regeneration to achieve highly active catalysts. The preparation of pure perovskite oxides or the integration of MOs requires long-term high temperature calcination (>800 °C), which results in a small specific surface area [50], and when used as a catalyst, only the outer surface is accessible. The introduction of porous structure will always increase the specific surface area of the material, thereby improving the catalytic performance. Since the catalytic activity is closely related to the surface properties of the material, the specificity of the perovskite structure makes it widely used in catalysis. Therefore, its application potential in solid propellants has also attracted attention.

Lanthanum ferrite (LaFeO_3_), a perovskite (ABO_3_) structure, has received extensive attention for its advantages such as stable framework structure and chemical stability [51]. Compared with single MOs, the special perovskite structure has more lattice distortions and vacancies [52].

Thirumalairajan et al. [53] synthesized the well-organized spherical LaFeO_3_ microspheres with an average diameter of 2–4 μm by a self-assembly process controlled synthesis using La(NO_3_)_3_·6H_2_O, Fe(NO_3_)_3_·6H_2_O and C_6_H_8_O_7_·H_2_O as starting materials.

The perovskite LaFeO_3_ nano-particles are able to catalyze the reaction between CO and NO_x_ in vehicle emissions [54,55]. It is also worth noting that CO and NO_x_ are the two main products of the thermal decomposition of HMX. The pressure exponent would decrease if the propellant for the reaction between CO and NO_x_ could be catalyzed on the combustion surface of nitrate plasticized polyether (NEPE) [56]. Therefore, perovskite-type oxides such as LaFeO_3_ are expected to be used as novel catalysts or modifiers for NEPE propellants.

Later, Wei et al. [57] used a novel stearic acid solution combustion method to directly prepare perovskite LaFeO_3_ and α-Fe_2_O_3_ with high specific surface area under the appropriate stearic acid-nitrate ratio of 1:1. The catalytic activity of the perovskite LaFeO_3_ and *α*-Fe_2_O_3_ for thermal decomposition of HMX was investigated by TG and TG-EGA techniques. The experimental results show that the catalytic activity of perovskite LaFeO_3_ is much higher than that of *α*-Fe_2_O_3_, which is due to the higher concentration of surface adsorbed oxygen (Oad) and hydroxyl groups of LaFeO_3_.

For solid propellants, Wei [57] and Wang [58] conducted a simple study on the application of LaFeO_3_ catalysts, but serious sintering problems occurred during the preparation of LaFeO_3_, which hindered the dispersion of LaFeO_3_ particles in the propellant matrix. Our group [59] prepared the spherical LaFeO_3_ with uniform particle size and good dispersion by solvothermal and post-heat treatment. However, due to the large particle size and low specific surface area of LaFeO_3_, its catalytic activity is limited to a certain extent. To improve its catalytic activity, a 3D core–shell heterostructure LaFeO_3_@MnO_2_ composite was constructed. The synthesis route is shown in Figure 8. The catalytic effect on different EMs of the prepared samples was studied by DSC, the curves are shown in Figure 9. Under the effect of LaFeO_3_@MnO_2_, the decomposition temperature of AP decreased from 403.73 °C to 281.38 °C, the energy release increased to 966.5 J·g^−1^ from 649.6 J·g^−1^, and the apparent activation energy of AP decreased from 139.05 kJ·mol^−1^ to 110.88 kJ·mol^−1^. In addition, LaFeO_3_@MnO_2_ also showed efficient catalysis for the thermal decomposition of CL-20 and HMX. The enhanced catalytic activity is attributed to the unique core–shell heterostructure and the synergistic effect between the LaFeO_3_ core and the ultrathin MnO_2_ shell.

### 3.2. Spinel Ferrite

Fe_2_O_3_, Fe_3_O_4_ with micro-nano structure and general ferrite compounds such as MOs·Fe_2_O_3_, M^II^Fe^III^_2_O_4_ and MFe_2_O_4_ (M is Mn, Co, Fe, Ni, Cu, etc.) play an important role in clean energy, green science and dealing with environmental pollution problems [60,61,62,63,64,65,66,67]. MOs containing iron, cobalt, and nickel elements usually have a spinel structure. The spinel-structure materials have special properties such as strong magnetic properties, high resistivity, high temperature resistance, and low preparation costs, which are widely used in magnetic recording materials, giant magnetic materials, microwave absorbing materials, etc. [68]. In addition, bimetallic oxides generally have better catalytic activity for the thermal decomposition of oxidants [69,70,71]. Several bimetallic iron oxides have been fabricated and used for thermal decomposition of AP [72,73,74,75,76,77,78]. These studies confirmed the catalytic activity of ferrate for the thermal decomposition of AP. However, due to the different APs that were used, the preparation method and the added amount of ferrate were quite different, it is hard to evaluate the catalytic performance of different ferrates based on data from different literatures. Therefore, we just list the data in Table 2.

Zhang et al. [79] investigated the catalytic effect of different ferrates on the thermal decomposition of AP. As shown in Figure 10, three ferrates (NiFe_2_O_4_, ZnFe_2_O_4_, and CoFe_2_O_4_) were prepared by a facile solvothermal method. The NiFe_2_O_4_, CoFe_2_O_4_ and ZnFe_2_O_4_ particles have hollow structures with average particle sizes of 70, 220 and 360 nm, respectively. The catalytic performance of ferrate on the thermal decomposition of AP was studied by DSC and TG-DTG methods, the corresponding curves are shown in Figure 11. The result indicated that CoFe_2_O_4_ had the best catalytic activity, which can decrease the T_HTD_ and *E*_a_ of AP by 108.99 °C and 37.38 kJ·mol^−1^, respectively. Further thermal analysis shows that the addition of ferrate had no obvious effect on the gas phase decomposition products of AP but had a greater impact on the energy barrier of AP pyrolysis. The excellent catalytic activity of CoFe_2_O_4_ is supposed to be attributed to the synergistic effect between Fe and Co.

To further enhance the effect, some preparation methods for enlarging the specific surface area were adopted. Xiong et al. [73] prepared nano CoFe_2_O_4_ with three-dimensional porous structure by a colloidal crystal template method using Co(NO_3_)_2_∙6H_2_O and Fe(NO_3_)_3_∙9H_2_O as the precursors. The prepared porous nano CoFe_2_O_4_ particles have a typical spinel structure with a pore size of about 200 nm (Figure 12). The specific surface area is significantly higher than that of 40 nm spherical CoFe_2_O_4_, reaching 55.646 m^2^∙g^−1^. The catalytic effects of porous nano CoFe_2_O_4_ and spherical nano CoFe_2_O_4_ on the thermal decomposition performance of AP were comparatively studied by DSC, and the DSC curves are shown in Figure 13. The porous nano CoFe_2_O_4_ with content of 6% can significantly reduce the HTD of AP by up to 91.46 °C. Moreover, the porous CoFe_2_O_4_ with only 2% can increase the decomposition reaction heat up to 1120.88 J∙g^−1^, which is 2.3 times that of pure AP. The thermal decomposition mechanism of AP catalyzed by porous CoFe_2_O_4_ was explored. Due to the high specific surface area (55.646 m^2^∙g^−1^) and high adsorption capacity of porous nanocomposite MOs, the formed gaseous intermediate products are adsorbed on the surface of porous nanocomposite CoFe_2_O_4_ (as shown in Figure 14). Then, the gaseous intermediate products are desorbed after electron transfer on the active site and separated from the pore wall to generate final products such as HCl, H_2_O, Cl_2_, O_2_, NO, N_2_O, and NO_2_ [80].

Wang et al. [81] used nickel nitrate, ferric nitrate, and cerium nitrate as oxidants, water-soluble hydrazine fuels as reducing agents, and added metal cation complexing agents and dispersants to prepare well dispersed Ce-doped nano NiFe_2_O_4_ by sol combustion synthesis method. The prepared nano powder has a particle size range of 30–60 nm, good dispersibility, and the Ce doping content is within 0.09 (as shown in Figure 15a). The catalytic effect of the product on the thermal decomposition of AP was studied by DSC (as show in Figure 15b). The results show that with the increase in Ce doping amount, the nano particles make the HTD of AP gradually decrease, and the apparent decomposition heat increases. When the Ce doping amount is 0.09, the HTD of AP is reduced by 57.8 °C.

Table 2 summarizes the related preparation methods of the pervoskite and spinedal ferrite as well as their catalytic effect to EMs thermal parameters in recent years. The catalytic effect of the ferrites is usually better than the corresponding MOs with single metal cation due to the synergistic effect of the bimetal.

**Table 2 molecules-28-02035-t002:** Summary of thermal behavior parameters of different ferrite EMs composites.

Materials	Samples	Preparation Method	Mass Ratios	*T* _p_	Δ*H*	*E*	Refs.
ABO_3_	AP	solvothermal and post-heat treatment	0	403.7	-	139.1	[59]
	LaFeO_3_/AP		20%	343.2	-	126.5	
	LaFeO_3_@MnO_2_/AP		20%	436.4	-	110.9	
	CL-20		0	252.1	-	-	
	LaFeO_3_/CL-20		20%	247.4	-	-	
	LaFeO_3_@MnO_2_/CL-20		20%	245.9	-	-	
	HMX		0	283.0	-	-	
	LaFeO_3_/HMX		20%	279.9	-	-	
	LaFeO_3_@MnO_2_/HMX		20%	273.4	-	-	
XY_2_O_4_	AP	colloidal crystal template	0	413.8	488.2	-	[73]
Nanoporous	CoFe_2_O_4_/AP		1%	354.5	950.5	-	
Nanoporous	CoFe_2_O_4_/AP		2%	356.0	1120.8	-	
Nanoporous	CoFe_2_O_4_/AP		3%	341.0	997.2	-	
Nanoporous	CoFe_2_O_4_/AP		4%	328.7	1109.6	-	
Nanoporous	CoFe_2_O_4_/AP		5%	329.3	1114.5	74.0	
Nanosphere	CoFe_2_O_4_/AP		5%	348.2	-	94.6	
Nanoporous	CoFe_2_O_4_/AP		6%	322.4	971.1	-	
Nanoporous	CoFe_2_O_4_/AP		7%	327.5	1113.2	-	
XY_2_O_4_	AP	solvothermal	0	404.3	-	162.3	[79]
	NiFe_2_O_4_/AP		10%	322.3	-	-	
	ZnFe_2_O_4_/AP		10%	373.4	-	-	
	CoFe_2_O_4_/AP		10%	295.3	-	124.9	
Ce doped NiFe_2_O_4_		sol-gel combustion synthesis	0	432.7	857.0	-	[82]
NiFe_2_O_4_	NiFe_2_O_4_/AP		5%	413.2	1144.0	-	
NiFe_2_Ce_0_._03_O_4_	NiFe_2_Ce_0_._03_O_4_/AP		5%	407.9	1147.0	-	
NiFe_2_Ce_0_._06_O_4_	NiFe_2_Ce_0_._06_O_4_/AP		5%	406.8	1190.0	-	
NiFe_2_Ce_0_._09_O_4_	NiFe_2_Ce_0_._09_O_4_/AP		5%	374.9	1259.0	-	

## 4. The Preparation of Carbon Composites and Their Application on the EMs

Nanoscale transition MOs used as combustion rate catalysts are proved to be able to effectively promote thermal decomposition of EMs and improve the combustion performance of EMs. Many transition metal oxides, such as CuO, NiO, MnO_2_, TiO_2_, and Fe_2_O_3_, have been used as burn rate catalysts for solid propellants. Among them, *α*-Fe_2_O_3_ is widely used due to its abundant, low cost, and nontoxic properties. However, the large surface area of this nanoscale catalyst always tends to agglomerate, which was one of the reasons for the reduced catalytic activity. In recent years, carbon materials such as graphene and its derivatives, carbon nanotubes, graphitic carbon nitride, etc., have been used as catalyst support and EMs due to its large specific surface area, excellent mechanical properties, good electrical and thermal conductivity.

### 4.1. Hematite/Graphene Composites

Since Novoselov obtained single layer graphene sheets by mechanical exfoliation in 2004 [83], graphene has developed rapidly in the fields of basic science and energy engineering. As a single atom thick graphite sheet with a two-dimensional honeycomb structure, it has a large theoretical specific surface area (2630 m^2^ g^−1^), high electrical conductivity, thermal conductivity and excellent mechanical properties [84]. These years, graphene has shown considerable potential in applications such as biomolecule detection, cell/tissue imaging, energy storage, catalysis, and batteries [82,85,86,87,88,89].

Graphene can affect the size and morphology of nanocrystalline materials and can make nanocrystals evenly distributed on the surface of graphene. It is worth noting that the synergistic effect between graphene and nanoparticles will always bring significantly enhanced performance. These advantages make the preparation and application of nanoparticle/graphene composites become a research hotspot [90,91,92,93,94]. In addition, graphene is also an excellent additive material for the construction of insensitive energetic components, which can be used to improve the safety performance of energetic components [95,96].

Qin et al. [97] used graphene oxide (GO) and ferric nitrate as raw materials to prepare Fe_2_O_3_/graphene composites by hydrothermal method. In brief, A certain amount of GO was added to deionized water, stirring for 1 h, and then ultrasonicating for 0.5 h to form a good GO suspension. After that, Fe(NO_3_)_3_·9H_2_O was put into a 250 mL three necked flask; then, the GO solution was added, and some aqueous ammonia solution was slowly added dropwise at room temperature until pH = 10–11. After stirring the solution for 2 h, 1 mL of hydrazine hydrate (98%) was added and evenly stirred. Then, the mixture was poured into a 20 mL hydrothermal reactor, reacted at 180 °C for 12 h, cooled to room temperature naturally after the reaction, finally the resulting colloidal product was washed and dried by centrifugation for several times.

Fe_2_O_3_/G/HMX was tested by DSC at different heating rates, and the effect of heating rate on the thermal decomposition characteristics of HMX and its apparent activation energy was studied.

Our group [98] synthesized two kinds of hematites with different morphologies (rod-like rFe_2_O_3_ and granular pFe_2_O_3_) by hydrothermal method. Then, the two composites (rFe_2_O_3_/G and pFe_2_O_3_/G) with three dimensional network structure were prepared by interfacial self-assembly method. The catalytic activity of the prepared Fe_2_O_3_ and Fe_2_O_3_/G for the thermal decomposition of CL-20 was investigated by DSC.

Figure 16a shows that pure CL-20 has exothermic behavior with peak temperature *T*_p_ = 525.28 K. Under the catalysis of rFe_2_O_3_/G, pFe_2_O_3_/G, rFe_2_O_3_ and pFe_2_O_3_, the thermal decomposition peak temperature of CL-20 were reduced by 8.34, 7.69, 6.28 and 5.84 K, respectively. The corresponding TG curves show that the mass loss rate reaches the maximum value at 515.04, 515.75, 516.87 and 517.82 K, which are all lower than pure CL-20 (*T*_p_ = 522.88 K; Figure 16b).

In addition, the catalytic effect rFe_2_O_3_/G towards HMX and RDX was also investigated. Figure 17 show that the exothermic decomposition temperatures of pure HMX and RDX at a heating rate of 10 K·min^−1^ are 556.31 K and 515.65 K, respectively. After mixing with the prepared catalysts, the decomposition process of HMX and RDX changed significantly. The distinct endothermic melting peak of pure HMX completely disappears after the addition of rFe_2_O_3_/G or rFe_2_O_3_. The initial decomposition temperature of rFe_2_O_3_/G/HMX was lower than that of rFe_2_O_3_/HMX, and the exothermic decomposition peak temperature decreased from 556.31 K to 551.72 K and 553.18 K, respectively. In Figure 17b, although the position of the endothermic temperature of RDX did not change, the exothermic peak changed significantly with the addition of rFe_2_O_3_ or rFe_2_O_3_/G. A sharp peak appeared in the DSC curve of rFe_2_O_3_/RDX, and the peak temperature was 507.23 K. Therefore, Fe_2_O_3_/G particles have stronger catalytic activity for the thermal decomposition of CL-20, HMX and RDX, comparing with the MOs alone. The reason lies in the synergistic effect between Fe_2_O_3_ and graphene and the high specific surface area of the Fe_2_O_3_/G composite.

Therefore, GO could be a good support material to decrease the aggregation degree of single MOs. Pei et al. [99] prepared Fe_2_O_3_ nano-particles through hydro-thermal method using FeCl_3_·6H_2_O and sodium acetate as raw materials. Then, Fe_2_O_3_/GO nanocomposites were synthesized by vacuum-freeze-drying. The procedure and the morphology of the Fe_2_O_3_ and Fe_2_O_3_/GO was shown in Figure 18. The shape of pure Fe_2_O_3_ nanoparticles is cubic (Figure 18b) and the average size of the prepared nanoparticles was 36.42 ± 6.47 nm. The Fe_2_O_3_ nanoparticles were well combined and wrapped by GO flakes (Figure 18d,e).

The catalytic performance of the prepared Fe_2_O_3_/GO nanocomposites for thermal decomposition of AP was investigated by DSC technique. Figure 19 show the DSC curves of pure AP and AP with 3 wt% ratio of different Fe_2_O_3_/GO nanocomposites. The ad-mixture catalyst could reduce the onset temperature of the HTD step and decrease the HTD temperature of AP by 77 °C, and the amount of heat released with the catalyst is twice that of pure AP. The reason for this good catalytic activity is because GO provides an ideal supporting substrate for the uniform distribution of Fe_2_O_3_ nanoparticles; thus, Fe_2_O_3_ can provide positive holes to accelerate the decomposition process.

Fertassi et al. [100] used ferric chloride hexahydrate (FeCl_3_·6H_2_O) as the precursor to prepare *α*-Fe_2_O_3_ nanoparticles by hydrothermal method. GO was synthesized using a modified Hummers method, and rGO were successfully obtained by thermal reduction of GO. *α*-Fe_2_O_3_/rGO nanocomposites were prepared using ex situ synthesis in the presence of *α*-Fe_2_O_3_ nanoparticles and GO solution. The catalytic activity towards the thermal decomposition of AP was investigated. DTA results show that the obtained nanomaterials help to improve the thermal decomposition of AP; specifically, the HTD of AP is reduced from 432 to 380 °C. In the presence of these nanomaterials, the activation energy was also significantly reduced from 129 to 80.33 kJ·mol^−1^.

The solvent used in the solvothermal method has a great influence on the structure and properties of the synthesized materials. Zhang et al. [101] prepared Fe_2_O_3_/rGO composites by a hydro-thermal method using six different solvents (distilled water, ethanol, N-methylpyrrolidone, ethyl acetate, n-butanol, and N,N-dimethylformamide). The specific synthesis route and catalytic mechanism of thermal decomposition of AP are shown in Figure 20. Fe(NO_3_)_3_ and GO suspension were sealed in the Teflon reactor for 24 h at 180 °C. After washing and drying, the Fe_2_O_3_/rGO composite was obtained. The SEM results showed that the Fe_2_O_3_ nanoparticles were successfully anchored on the graphene surface (Figure 21a–e). However, the morphology of the composites and the dispersion of Fe_2_O_3_ nanoparticles were found to be significantly different with different solvents. Compared with the Fe_2_O_3_/rGO composites prepared in NBA, H_2_O, NMP, EA and EG solvents, the composites obtained in DMF solvent exhibited a thinner structure. Equally importantly, the distribution of Fe_2_O_3_ nanoparticles immobilized on graphene was also high using DMF as the reaction solvent, confirming that graphene can effectively prevent the aggregation of Fe_2_O_3_ nanoparticles. GO and rGO can significantly contribute to the better dispersion properties of nano particles [102,103].

In the presence of Fe_2_O_3_/rGO (DMF) composites, the HTD and apparent activation energy of AP decreased by 119.6 °C and 173.3 kJ·mol^−1^, respectively.

Atomic Layer Deposition (ALD) is a thin film coating technique that enables the production of nanometer films or nanoparticles in a highly controlled manner. Yan et al. [104] prepared rGO@Fe_2_O_3_ with finely dispersed Fe_2_O_3_ nanoparticles using the ALD route. Fe_2_O_3_ nanoparticles with uniform particle size are uniformly anchored to rGO nanosheets through surface chemical interactions, thus effectively suppressing the aggregation of Fe_2_O_3_ nanoparticles, and graphene nanosheets have high specific surface area and excellent electrical conductivity, which can increase the accessibility number of active sites and enhance electron transfer during catalysis. The morphologies of the prepared rGO@Fe_2_O_3_ was show in Figure 22a,b. DSC results in Figure 22c indicate that the presence of 5 wt% rGO@Fe_2_O_3_ composite makes the two decomposition stages of pristine AP almost merge into a single exothermic peak between the LTD and HTD stages. The rGO@Fe_2_O_3_ with 212 wt% of Fe_2_O_3_ on the rGO bases exhibits a significantly lower decomposition temperature compared with those of rGO and rGO@ Fe_2_O_3_ composites.

There are some other methods for preparing the Fe_2_O_3_/rGO composite. Elbasuney et al. [105] effectively prepared Fe_2_O_3_/rGO nanocomposites using a solution co-precipitation method in deionized water. After adding a hydrazine reducing agent, washing, centrifuging and drying, the composite nano particles were developed via calcinations at 400 °C. The TEM images of Fe_2_O_3_/rGO show that the hybrid nanocomposites consist of two-dimensional rGO sheets and were decorated with Fe_2_O_3_ nanocrystal. The catalytic performance of the Fe_2_O_3_/rGO nanocomposite towards the thermal behavior of AP was evaluated using DSC and TGA. The initial endothermic decomposition of AP is reduced by 16% and the total heat release was improved by 83%. The combination of rGO and Fe_2_O_3_ provides a high interfacial surface area, which can ensure the absorption of gaseous products on the catalyst surface with a surge in the total heat release.

### 4.2. Ferrite/Graphene Composites

Ferrite-based burn rate modifiers contain two metal elements, which may have catalytic effects on both elements. Carbon materials have been widely used as additives in solid propellants. Different methods are used to load ferrite nanoparticles on the surface of carbon materials, so that the catalyst has the advantage of both bimetallic elements and carbon materials. This type of composite material has the effect of reducing the migration ability of nanoparticles, which is of great significance to the long-term storage stability of propellants.

Xu et al. [106] prepared bismuth ferrite/graphene (BiFeO_3_/rGO) nanocomposites by a hydrothermal method using graphene oxide as a carrier. The effect of different addition amounts of BiFeO_3_/rGO nanocomposites on the thermal decomposition properties of AP was studied by differential thermal analysis. In order to study the effect of catalyst content on the thermal decomposition performance of AP, 1 wt%, 2 wt%, 3 wt%, 4 wt% and 5 wt% of BiFeO_3_/GO nanocomposites were added to AP, respectively. The results show that the BiFeO_3_/rGO nanocomposite has a good ability to catalyze the thermal decomposition of AP. When the mass fraction of BiFeO_3_/rGO nanocomposite is 4 wt%, the thermal decomposition catalytic performance is the best (reduce the HTD by 167 °C). The catalytic activity of BiFeO_3_/rGO nanocomposite was believed to accelerate the thermal decomposition reaction of AP. The high catalytic effect benefits from its nano-sized morphology and large specific surface area, which can provide more active reaction sites, and can fully react with AP.

The catalytic effect of bismuth ferrite is different from iron oxide, and obviously also different from the mixture of iron oxide and bismuth oxide. Guo et al. [107] used the hydrothermal method to prepare BiFeO_3_/GO, K and Ce doped BiFeO_3_/GO. The effect of doping on the particle size of bismuth ferrite was further confirmed by TEM (Figure 23a,b). The nanoparticles in K/BiFeO_3_/GO are 10–100 nanometers, and the Ce/BiFeO_3_/GO sample presents a large number of tiny particles with a size less than 20 nm. To study the catalytic thermal decomposition ability of the catalyst to the high energy explosive HMX, the catalyst and HMX were ground and mixed at a mass ratio of 1:4. The results are shown in Figure 23c,d, BiFeO_3_/GO doped with cerium ions showed a good catalytic effect on the thermal decomposition of HMX, and the onset temperature of thermal decomposition was advanced from 281 °C to 209 °C. Since Ce/BiFeO_3_/GO directly decomposes HMX in the solid phase, the initial decomposition temperature is greatly increased, indicating that it has a higher catalytic thermal decomposition activity than K/BiFeO_3_/GO.

The improved catalytic activity due to the synergistic effect of nanoparticles and rGO, which may broaden new applications in modifying the combustion behavior of AP-based composite propellants. Chen et al. [108] successfully prepared CoFe_2_O_4_/rGO hybrids by a facile one-pot solvothermal method, and their catalytic behavior for AP decomposition was investigated by varying their content in the range of 1–5% by differential thermal analysis. It was confirmed by TEM Figure 24a,b that RGO can effectively prevent the aggregation of CoFe_2_O_4_ nanoparticles. The synthesized CoFe_2_O_4_/rGO hybrid exhibited high catalytic activity for the thermal decomposition of AP, as shown in Figure 24c,d. The HTD temperature gradually decreased with the increase in the catalyst. After adding a 3% CoFe_2_O_4_/rGO hybrid, the HTD temperature of AP decreased by 104.7 °C, which was lower than that of bare CoFe_2_O_4_ with the same proportion. Thus, rGO can enhance the catalytic activity of CoFe_2_O_4_ and facilitate the thermal decomposition process. The improved catalytic activity may broaden new applications in modifying the combustion behavior of AP-based composite propellants.

They also successfully synthesized NiFe_2_O_4_/rGO nanoparticles by the solvothermal method [109]. The catalytic activity of the synthesized NiFe_2_O_4_/rGO nanoparticles on AP was characterized by DSC. The DSC curves show that with the addition of NiFe_2_O_4_/rGO nanoparticles, the LTD and the HTD merge into a single exothermic process, but the phase transition temperature position of AP still exists as AP. The HTD temperature decreases gradually with the increase content of the catalyst. Compared with pure AP, the HTD temperature of AP decreased by 103.8 °C when adding 5% NiFe_2_O_4_/rGO.

Wang et al. [110] also prepared ZnFe_2_O_4_/rGO nanohybrids by a facile one-pot hydrothermal method. The TEM images confirmed that the ZnFe_2_O_4_ NPs with a size of about 10 nm were anchored uniformly on the rGO sheet. DSC results indicated that ZnFe_2_O_4_/rGO nanohybrids exhibited enhanced catalytic activity for the thermal decomposition of AP compared with ZnFe_2_O_4_ NPs. The composites can reduce the secondary decomposition temperature of AP by 42.7 °C, 55.0 °C and 68.1 °C with the increase in the proportion (1%, 3%, 5%), respectively. Furthermore, the composite with the proportion of ZnFe_2_O_4_/rGO reduces the apparent activation energy of AP from 160.2 kJ mol^−1^ to 103.8 kJ mol^−1^. rGO is supposed to accelerate the electron flow, which can further accelerate the heterogeneous decomposition reaction rate of gas-phase molecules on the surface of NPs.

### 4.3. Carbon Nanotubes

Carbon nanotubes (CNTs) have nano-scale lumen structure, large specific surface area, surface energy and surface binding energy, and graphite like multi-layer tube wall structure, which can adsorb and fill particles. Moreover, with good chemical stability, good electrical conductivity and high mechanical strength, loading nanocatalysts on CNTs can improve the dispersion problem of nanoparticles and promote the transfer of electrons during the reaction, increase the catalytic effect, which makes it have a good application prospect in the catalyst carrier [111,112]. CNTs, as the adsorptive carrier for Fe_2_O_3_, can maximize the specific surface area of the prepared nanoscale Fe_2_O_3_ catalysts, thereby improving its catalytic efficiency.

Wang et al. [113] used the liquid-phase precipitation to prepare Fe_2_O_3_/CNTs composites. Additionally, the Fe_2_O_3_/CNTs composite nanocatalyst was found to be able to significantly reduce the decomposition temperature of AP and improve the burning rate of AP unit propellant. The catalytic activity of Fe_2_O_3_/CNTs composite for AP is better than that of nano-Fe_2_O_3_ and the simple mixture of nano-Fe_2_O_3_ and CNTs.

Jiang et al. [114] used purified CNTs as a carrier to prepare Fe_2_O_3_/CNTs composite particles by a sol-gel method. Figure 25a,b shows the SEM image of Fe_2_O_3_/CNTs and the DSC curves of the composite with a different ratio. The characterization indicated that the Fe_2_O_3_ nano-particles stuck on the surface of CNTs. Fe_2_O_3_/CNTs composite particles with a molar ratio of 1:7 can apparently decrease the thermal decomposition peak temperature of AP to 345.8 °C, reducing the apparent activation energy of AP from 178 kJ·mol^−1^ to 118 kJ·mol^−1^.

### 4.4. Graphitic Carbon Nitride

Graphitic carbon nitride (g-C_3_N_4_) has attracted much attention as a new generation of green non-metallic two-dimensional catalysts. In the typical structure of g-C_3_N_4_, C and N atoms form a highly delocalized large π-conjugated system by sp2 hybridization, becoming the lowest energy but most stable carbon nitride allotrope [116]. There are some reports related to the good photoelectrochemical catalytic performance [117,118,119], which indicates that the g-C_3_N_4_ does much contribution to reducing the bandgap, escalating the specific area of the made composite. In the EMs field, it was reported that adding 10% g-C_3_N_4_ to AP can advance the pyrolysis peak temperature of AP by 70 °C [120].

Wang et al. [121] prepared Fe_2_O_3_/g-C_3_N_4_ composite using FeCl_3_·6H_2_O and melamine as the precursors by sol-gel method. The SEM image indicates that Fe_2_O_3_ particles are tightly supported by g-C_3_N_4_. The catalytic effect of the prepared composite on the thermal decomposition of AP was investigated by DSC. DSC tests showed that the addition of the composite catalysts with 5% mass fractions reduced the HTD and LTD of AP to 348.1 °C and 281.7 °C, respectively.

The properties of ferrite nano particles with g-C_3_N_4_ were studied by some researchers. Wan et al. [122] self-assembly combined CoFe_2_O_4_ with a prepared g-C_3_N_4_ using Fe(NO_3_)_3_·9H_2_O and Co(NO_3_)_2_·6H_2_O as precursors in a hydrothermal environment. DSC results show (Figure 26) that CoFe_2_O_4_/g-C_3_N_4_ reduces the thermal decomposition peak temperature of HMX and TKX-50 by 7.0 °C and 41.3 °C, respectively, and the apparent activation energy decreases by 341.1 kJ·mol^−1^ and 21.0 kJ·mol^−1^.The as-prepared composite have a better catalytic effect on HMX and HATO(TKX-50). Now some other literatures [123] involved g-C_3_N_4_ as a carrier show that this carbon material exhibits good catalytic effect on the thermal decomposition of EMs. Thus, g-C_3_N_4_ may be a prospective application as a combustion catalysis.

Table 3 summarizes the related preparation methods and corresponding thermal parameters in recent years. Carbon materials, especially graphene, contribute much to the enhancement of the catalytic effect of EMs due to the synergistic effect of NPs and rGO.

## 5. The Nano-Thermites and Their Application

For the traditional micro-thermite system, there are many unfavorable factors: large component size, uneven mixing, high ignition temperature (>900 °C), slow burning rate and incomplete burning, while when the particle components are nanosized, the contact area and the degree of bonding between the components can be greatly increased.

Nano-thermite is defined as a metastable intermixed composites (MICs) of nano-aluminum powder and nano-metal oxide [124,125]. After compounding in a certain proportion, under the condition of heating or mechanical impact, the violent redox reaction happened, and a large amount of heat released. The early thermite was mainly prepared from Al powder and different MOs in a certain ratio, and it was ignited to generate Al_2_O_3_ and metals to release a lot of heat. With the development of technology, in addition to aluminum powder, some other active metals such as magnesium, lithium, nickel and boron can also be mixed with metal oxides, and a similar aluminothermic reaction occurs.

At present, nano-thermite is widely used in the fields of pyrotechnics, explosives and rocket propellants. Usually nano-thermite used in solid propellants can improve its specific impulse and combustion efficiency and enhance mechanical properties. Studies have shown that in composite propellants, when nano-aluminum powder replaces ordinary aluminum powder, the propellant burning rate can be increased by 2 to 5 times. Compared with ordinary ignition powder, the biggest advantage of nano-thermite modified ignition powder is that it can generate a lot of heat during the reaction and can reduce the electrostatic discharge sensitivity of nano-thermite ignition powder. In addition, the compounded composite materials can change the agglomeration of the individual nano-aluminum powder; at the same time, the compounded nano-thermite can ensure the active content and increase the contact area so as to improve the combustion performance of solid propellants [126,127,128,129,130,131].

### 5.1. Influence of Fuel Additives on Nano-Thermite

The traditional thermite always uses aluminum powder as fuel; however, the exothermic performance sometimes cannot meet the current industrial production. Adding some fuel additives can improve the heat release of the Al/Fe_2_O_3_ thermite.

Nie et al. [132] prepared bimetallic thermite powder (Al/Fe_2_O_3_/Ni) by an ultrasonic physical mixing method. They carried out experiments about the aged samples at two relative humidity (RH) levels in air under an isothermal environment (~60 °C). The measured total energy output of the aged and fresh samples with different nickel contents indicated that the energy output in powders with a moderate humidity level of 20% was comparable across the nickel range within 20%. Under extreme humidity conditions of 75%, the energy output of the Ni-free Al/Fe_2_O_3_ nanothermite powder was significantly reduced by 33%. Thus, Ni can reduce the sensitivity of nano-thermite powder to humidity at higher temperature, which can prolong the shelf life of the thermite powder.

Shen et al. [133] successfully prepared Al/B/Fe_2_O_3_ nano-thermite by the sol-gel method using ultrasonic under mild and non-toxic conditions. The characterizations indicated that nano-aluminum and micro-boron were uniformly dispersed in the pores of the iron oxide gel. DSC tests show that the temperature of the exothermic reaction heat of Al/B/Fe_2_O_3_ nanothermites shifts to a lower temperature and the heat release obtained by the sol-gel approach is more than that of the simply physical mixture. Boron was also found to have the highest volumetric heat of combustion, so it is a good choice as a fuel additive in thermite.

Recently, our group [134] prepared bismuth cation-doped hematite (Bi-Fe_2_O_3_) with different Bi content by a one-step hydrothermal method and used it as a thermal decomposition catalyst for EMs and oxidants in Al-based thermite (Al/Bi-Fe_2_O_3_). SEM images (Figure 27a–e) show a three-dimensional flower-structure when doping with different molar ratio of Bi/Fe is 0.2 or 0.3. The aluminothermic reaction was evaluated by DSC in the temperature range of 40–900 °C, as shown in Figure 27f. The thermite reaction of Al/Bi-Fe_2_O_3_ is exothermic with the peak temperatures appearing at 583–739 °C. With the increase in Bi, the temperature of the first exothermic peak of Al/Bi_x_-Fe_2_O_3_ hybrids is delayed by 6–15 °C, but their second exothermic peak temperature decreases by 58–101 °C. However, the energy release is lower than Al/Fe_2_O_3_ and decreases with the increasing doping content of Bi. Al/Bi-Fe_2_O_3_ showed a shorter ignition delay time than Al/Fe_2_O_3_.

The combustion process of the Al/Bi-Fe_2_O_3_ thermites with different Bi doping content was tested by CO_2_ continuous laser and recorded by a high-speed camera (Figure 28). With the increase in the Bi doping content, the ignition delay time decreases and Bi_0_._3_-Fe_2_O_3_ can minimize the delay time of Fe_2_O_3_ most.

### 5.2. Influence of Oxidant and Structure on Nano-Thermite

The combustion of nano-thermite is a complex multiphase reaction process, including heat conduction, convection, feedback and radiation. Usually, uniform distribution and closeness means close interfacial contact and short mass transfer and heat diffusion distances between components, which will contribute to promoting the reaction kinetics and improving the combustion performance of the thermite. Thus, various interface control techniques, such as atomic layer deposition [135], self-assembly [136], sol-gel synthesis [137], electrophoretic deposition [138], in situ growth [139], and arrested reactive milling [140], were used.

The type of oxidant in the thermite is one of the important factors in the exothermic, ignition and combustion properties of composites [141,142]. In the past few decades, many studies of thermite have focused on the preparation and performance improvement of Al/CuO, Al/Fe_2_O_3_, Al/Bi_2_O_3_, Al/NiO, Al/WO_3_ MICs, in which the oxidants mainly focus on metal oxidation objects [141,142,143,144,145,146]. Wu et al. [145] studied the thermal behavior of four commonly used thermite systems in different atmospheres: Al/CuO, Al/Fe_2_O_3_, Al/Fe_3_O_4_, and Al/Co_3_O_4_. The DSC curve is shown in Figure 29. The energy release before 600 °C was found to be much greater in air than in argon. The system of Al/CuO nanothermite exhibits the highest heat release with 5125 J·g^−1^, while Al/Co_3_O_4_ releases 3958 J·g^−1^ heat, which is the lowest among the four samples.

Therefore, the nature of the oxidant has an essential influence on the reaction kinetics and reaction mechanism of the thermite composites. Composite MOs could combine the properties of two single MOs with special properties such as electrical conductivity, excellent catalytic activity, and synergistic effect, which may be better than the mixture of two MOs [147,148].

As far as the structure of the composite material is concerned, the uniform distribution and compactness mean that the interfacial contact between the components is close, and the mass transfer and thermal diffusion distances are short, which is very beneficial for promoting the reaction kinetics and improving the performance.

Wang et al. [149] assembled a novel NC coated aluminum/copper ferrite (Al/CuFe_2_O_4_@NC) thermite using electrospray technique to improve the distribution and enhance the interfacial contact between the components. The thermites with these different oxides were obtained using NC as a binder. The thermal behavior, laser ignition and combustion properties of the assembled Al/CuFe_2_O_4_@NC were evaluated and compared with their physical mixtures. Compared with Al/CuO@NC and Al/Fe_2_O_3_@NC composites, Al/CuFe_2_O_4_@NC composites have lower boost rate and longer duration. Furthermore, the thermites prepared by electrospraying exhibit better properties than the samples by simple physical mixing. Thus, uniform distribution and close interfacial contact of the fuel and the oxidant do much help in enhancing the performance.

Further combustion pressure experiment was carried out for exploring the pressurization characteristics of the thermite during the combustion process. Representative time resolved pressure and light emission curves of Al/CuO@NC, Al/Fe_2_O_3_@NC and Al/CuFe_2_O_4_@NC composites are shown in Figure 30a,b, respectively. Light emission represents the duration of recombination combustion in the confined combustion chamber. The result indicated that Al/CuO@NC has the largest pressurization rate and the shortest duration, indicating a vigorous combustion reaction and a high reactivity. As a contrast, the Al/CuFe_2_O_4_@NC composite has a lower pressing rate and longer duration. Thus, the oxidant can greatly affect the combustion duration and pressurization rate of the composite.

Shi et al. [150] first prepared monodispersed polystyrene (PS) by an emulsifier-free polymerization method. Then, the macroporous (3DOM) NiFe_2_O_4_ was obtained through immersing the colloidal crystal template to the methanol and ethylene glycol solution (volume ration of 1:1) of Fe(NO_3_)_3_·9H_2_O and Ni(NO_3_)_2_·6H_2_O with stirring and the later calcination. The Al/NiFe_2_O_4_ nanothermite was finally prepared by magnetron sputtering nano-Al to the 3D-DOM NiFe_2_O_4_ under vacuum. The surface view of the PS template, the honeycomb structure NiFe_2_O_4_ films and the Al/NiFe_2_O_4_ were shown in Figure 31. The unique three-dimensional pore structure helps to increase the heat release of the thermite reaction and makes the as-prepared thermite to be easily ignited by laser.

### 5.3. Nano-Thermites with Different Morphologies and Their Catalytic Effects on EMs

Al/Fe_2_O_3_ is a traditional thermite with an energy density of 3.71 kJ∙g^−1^ and a reaction temperature of up to 3135 K. It is widely used as propellants and additives for high explosives, airbag ignition materials, welding torches, etc. Fe_2_O_3_ with different morphologies presents differently in the thermite reaction.

Our group [8] studied the thermite reaction of Al/Fe_2_O_3_ with three different morphologies which is granular, oval and polyhedral of Fe_2_O_3_. The SEM images and the DSC curve are shown in Figure 32. The thermite reaction happened before the melting of Al powder when the particle size is micro or nano. Additionally, the thermal process was quite different among the three thermites. The Al/Fe_2_O_3_ (granular) particles exhibit one rapid exothermic process under 660 °C, which may attribute to the highest burning rate in AP/HTPB composite propellants.

Yang et al. [151] successfully used the electrospray process to prepare aluminum nanopowder and iron oxide nanopowder Al/Fe_2_O_3_/RDX/NC composites with different RDX contents. The RDX in the precursor solution plays an important role in the morphology of the composites. After adding RDX to the composite, a loose structure was formed. DSC curves (Figure 33) showed that the decomposition temperature of RDX was reduced by about 20 °C through the electrospray prepared composition compared to RDX. The combustion performance in air showed that RDX reduced the combustion performance of nano-thermite, and the combustion intensity decreased with the increase of RDX content. In addition, the combustion chamber test shows that the maximum pressure peak of the Al/Fe_2_O_3_/RDX/NC composite is greatly improved.

The addition of AP to the porous network of Al/Fe_2_O_3_ can generate MICs with uniform density. Gao et al. [152] immersed nano-aluminum particles into the pores of hematite matrix to form Al/Fe_2_O_3_ nano-thermite through sol-gel processing. Then, porous Al/Fe_2_O_3_ particles were combined with AP using wet impregnation and solvent-antisolvent techniques. Thus, the MICs-AP/Al/Fe_2_O_3_ nano-thermite was obtained. The SEM images (Figure 34a) show that AP and nano-aluminum are dispersed in the pores of the hematite gel with a large specific surface area. DSC curves of the AP, AP + Al/Fe_2_O_3_ and AP/Al/Fe_2_O_3_ were plotted, as shown in Figure 34b. The Al/Fe_2_O_3_ nano-thermite plays a catalytic role in the thermal decomposition of AP, and the interaction of the thermite reaction is greatly enhanced by the accelerated decomposition of AP. The activation energy of AP was decreased from 176.84 kJ·mol^−1^ to 109.22 kJ·mol^−1^.

As the main energetic component, RDX is widely used in solid propulsion. However, the influence of nano-thermite Al/Fe_2_O_3_ on the thermal decomposition characteristics of RDX is rarely studied. Our group [153] prepared hollow short rod-like nano-Fe_2_O_3_ by the hydrothermal method and combined it with nano-Al particles by an ultrasonic dispersion method to prepare super thermite. The SEM images of the rod-like Fe_2_O_3_ and Al/Fe_2_O_3_ are shown in Figure 35a,b. The effects of the super thermite Al/Fe_2_O_3_ on the thermal decomposition properties of RDX were studied by DSC (Figure 35c). The results show that the addition of nano-thermite changes the thermal decomposition process of RDX and intensifies the secondary gas phase reaction of RDX. With the increase in super thermite content, the peak shape of the decomposition peak of RDX significantly changes. The second shoulder peak temperature for the thermal decomposition peak of RDX decreases.

Furthermore, our group [154] used FeCl_3_·6H_2_O and NaOH as raw materials to prepare rod-like Fe_2_O_3_ by a hydrothermal synthesis method at 180 °C for 6 h. Another two Fe_2_O_3_ nano-particles with olivary and polyhedral morphologies were also prepared. Then, Al/Fe_2_O_3_/NC composites were designed and fabricated using various morphologies of Fe_2_O_3_. The TEM images of the three samples with different morphologies were shown in Figure 36. The effect of Al/Fe_2_O_3_ on the thermal decomposition properties of NC was studied in detail by DSC (Figure 37). Compared with other morphological samples, the performance of Al/Fe_2_O_3_ containing rod-like Fe_2_O_3_ is greatly improved. The *E*_a_ and thermal ignition temperature (*T*_beo_) of Al/r-Fe_2_O_3_/NC are the lowest.

Recently, 3D printing techniques raise much interest in composite preparation. MICs inks for thermite is a challenging work. Zhong et al. [155] used a core–shell nozzle for 3D printing to prepare Al/Fe_2_O_3_/F2311 thermite using 15% (wt%) fluororubber (F2311) as polymer binder (Figure 38a). The 3D printing technique ensures that the formulations all have hollow uniform filaments with a constant diameter of ~0.5 mm. DSC results (Figure 38b) show that the onset thermal decomposition temperature of Al/Fe_2_O_3_/F2311 is 368 °C, and the heat release is 1552 J·g^−1^.

Table 4 summarizes the relevant parameters such as their preparation methods and applications of thermite. Al is the most commonly used fuel in composite solid propellants, the oxidizing agent in the aluminothermic composite has a significant effect on the energy release, ignition delay and combustion performance. The uniform distribution and tight interfacial contact between fuel and oxidant is beneficial to shorten the distance for mass transfer and thermal diffusion, thus enhancing the performance of thermites.

Different kinds of thermites like Al/KMnO_4_, Al/SnO and Al/AgIO_3_ are prepared to meet different applications. The fabrication methods include some new techniques such as vapor deposition, high-energy ball milling, electrophoretic deposition, etc. [156]. With the development of preparation methods and techniques, the thermite will be more specific and multifunctional.

## 6. Future Outlook

The preparation of hematite with different morphologies and their enhanced effect on the thermal decomposition of AP, RDX, HMX, CL-20, etc., were summarized. The morphology has been proved to be able to greatly affect the thermal decomposition behavior of EMs. Making ferrite, combined with some carbon materials, can promote the catalytic effect of hematite because more active spots were supplied. GO was a good carrier of the catalyst which can effectively decrease the aggregated degree of the nano metal oxide particles. The g-C_3_N_4_ raised much attention as a two-dimension carbon material. When combining hematite with nano-sized Al powder to make the super-thermite, the composite can release more energy and exhibit good catalytic effects on EMs. With the development of material and preparation techniques, more new hematite formulations will be explored, the performances of EMs will be enhanced. The application of hematite in EMs still needs long term research and exploration. The combination of carbon materials with high specific surface area or porous structure can greatly help to solve the problems of poor dispersion in thermite preparation and energetic composites. Thereby enhancing the properties of hematite or hematite composites.

## Figures and Tables

**Figure 1 molecules-28-02035-f001:**
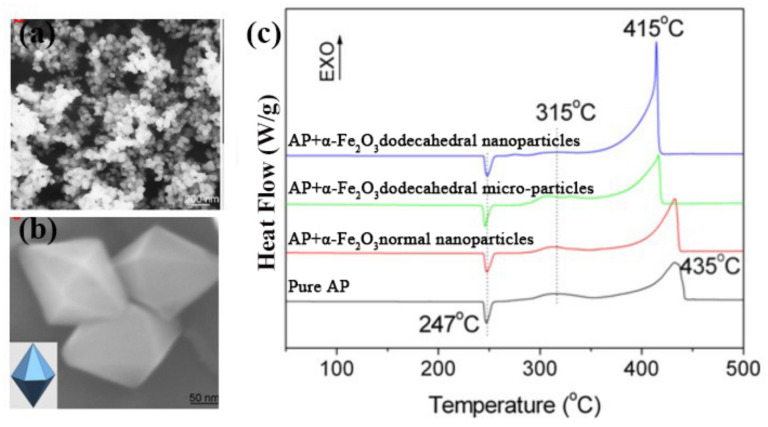
SEM images of (**a**) synthesized product; The inset of (**b**) is a geometrical model of the dodecahedral particle; (**c**) DSC curves for pure AP, AP+*α*-Fe_2_O_3_ dodecahedral nanoparticles, AP+*α*-Fe_2_O_3_ normal nanoparticles, AP+*α*-Fe_2_O_3_ dodecahedral micro-particles. (**a**–**c**) Reproduced with the permission of ref. [35]. Copyright 2014, Journal of Nanoparticle Research.

**Figure 2 molecules-28-02035-f002:**
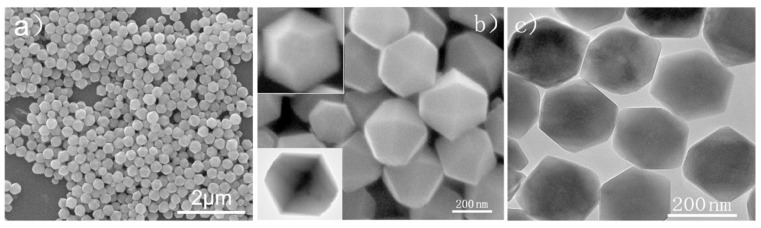
The tetrakaidekahedral *α*-Fe_2_O_3_ nanocrystals synthesized at 200 °C for 6 h. (**a**,**b**) Low- and high-magnification SEM image of tetrakaidekahedral *α*-Fe_2_O_3_ crystals. Insets (top left and bottom left) represents the cross-section of the as-prepared product and HRTEM images of a single tetrakaidekahedral crystal; (**c**) TEM image of the tetrakaidekahedral *α*-Fe_2_O_3_ crystals. (**a**–**c**) Reproduced with the permission of ref. [37]. Copyright 2011, Inorganic Chemistry.

**Figure 3 molecules-28-02035-f003:**
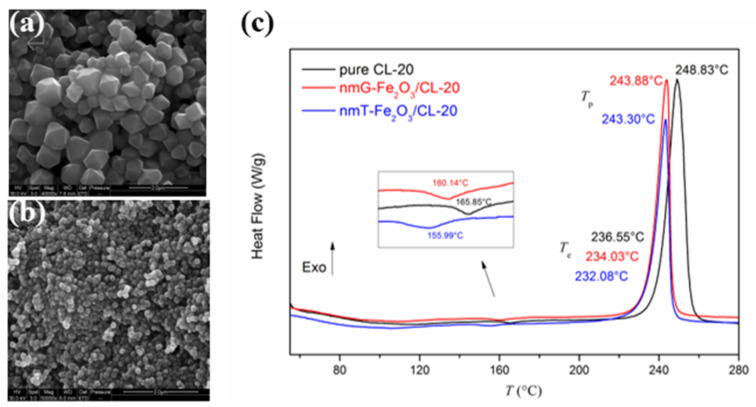
SEM images of the two nano-Fe_2_O_3_ with a resolution of 2 μm. (**a**) tetrakaidekahedral -Fe_2_O_3_ (nmT-Fe_2_O_3_), (**b**) grainy-Fe_2_O_3_ (nmG-Fe_2_O_3_) and DSC curves of CL-20, nmT-Fe_2_O_3_/CL-20 and nmG-Fe_2_O_3_/CL-20 at a heating rate of 10 K·min^−1^ (**c**). (**a**–**c**) Reproduced with the permission of ref. [38]. Copyright 2016, Propellants Explosives Pyrotechnics.

**Figure 4 molecules-28-02035-f004:**
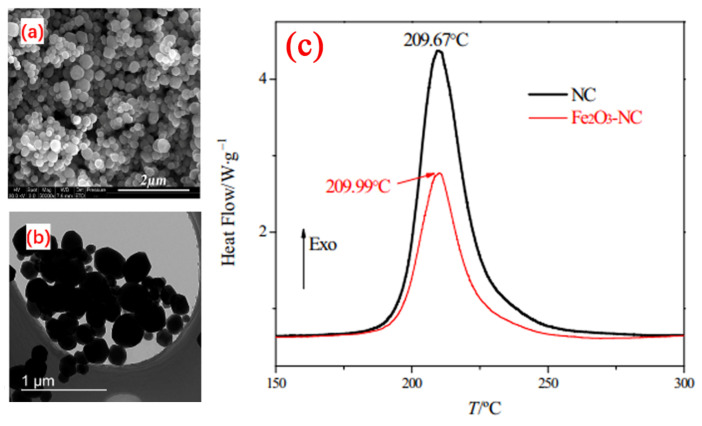
SEM (**a**), Low-magnification TEM (**b**) micrographs of α-Fe_2_O_3_ nanoparticles and DSC curves of NC and Fe_2_O_3_-NC (**c**) obtained at a heating rate of 10 °C min^–1^. (**a**–**c**) Reproduced with the permission of ref. [40]. Copyright 2016, Journal of Analytical and Applied Pyrolysis.

**Figure 5 molecules-28-02035-f005:**
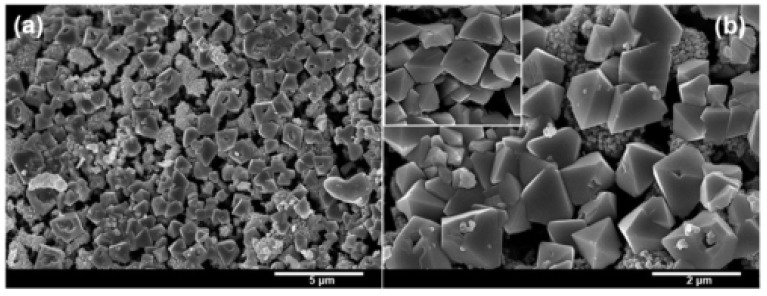
FESEM images at low (**a**) and high (**b**) magnification of the synthesized Fe_2_O_3_ hexagonal cones. (**a**,**b**) Reproduced with the permission of ref. [45]. Copyright 2015, New Journal of Chemistry.

**Figure 6 molecules-28-02035-f006:**
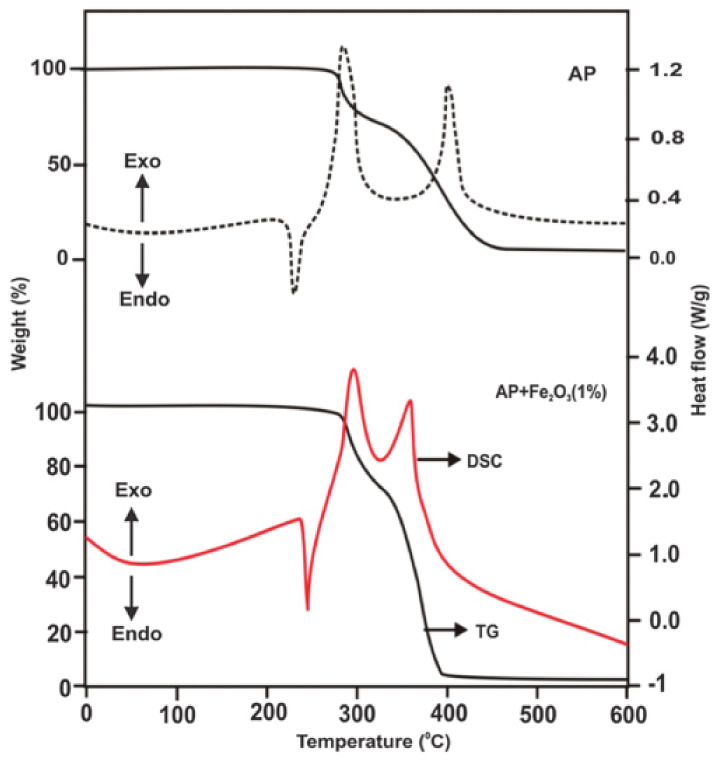
TG-DSC curves of pure AP and AP with the green synthesized Fe_2_O_3_ hexagonal cones (1 wt%). Reproduced with the permission of ref. [45]. Copyright 2015, New Journal of Chemistry.

**Figure 7 molecules-28-02035-f007:**
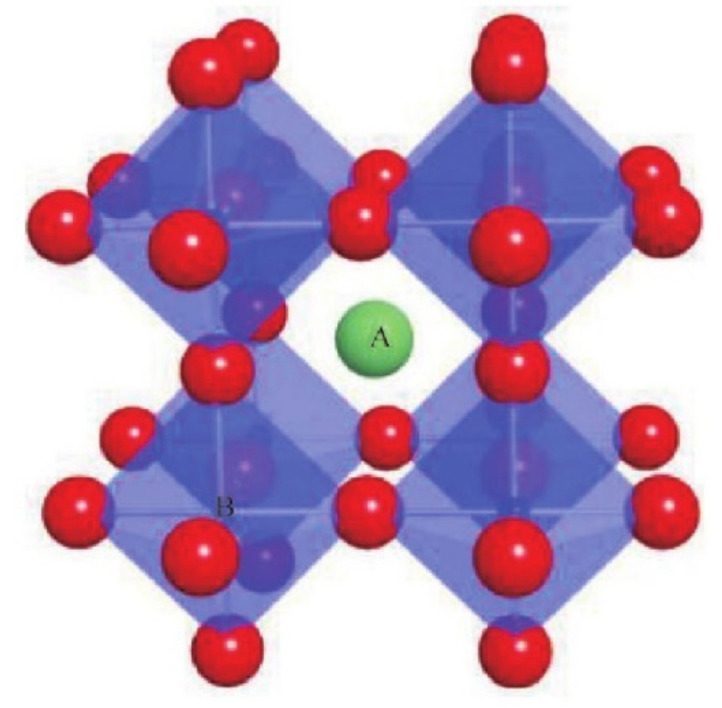
Perovskite crystal structure. Reproduced with the permission of ref. [47]. Copyright 2021, Chemical Industry and Engineering Progress.

**Figure 8 molecules-28-02035-f008:**
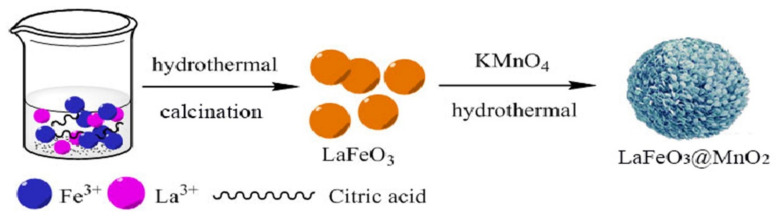
Synthetic routes of LaFeO_3_ and LaFeO_3_@MnO_2_. Reproduced with the permission of ref. [59]. Copyright 2020, Advanced Powder Technology.

**Figure 9 molecules-28-02035-f009:**
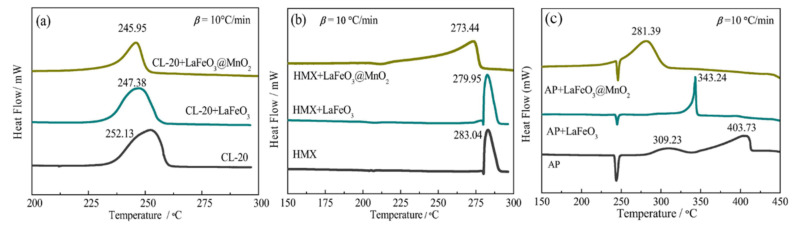
DSC curves for (**a**) CL-20 and (**b**) HMX (**c**) AP with different catalysts. (**a**–**c**) Reproduced with the permission of ref. [59]. Copyright 2020, Advanced Powder Technology.

**Figure 10 molecules-28-02035-f010:**
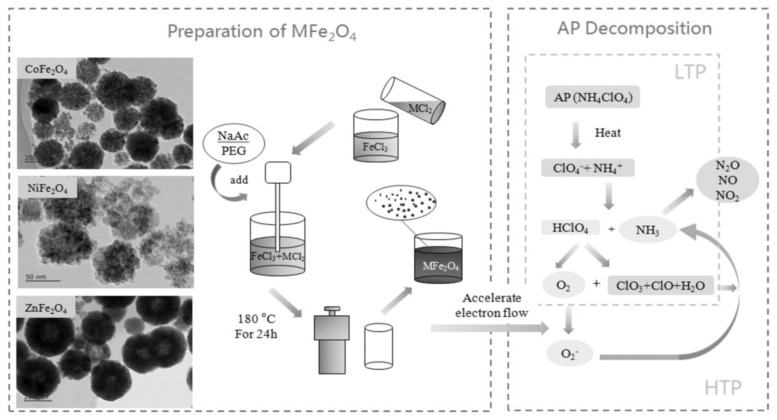
Illustration of ferrate fabrication and the catalytic mechanism for thermal decomposition of AP. Reproduced with the permission of ref. [79]. Copyright 2020, Propellants Explosives Pyrotechnics.

**Figure 11 molecules-28-02035-f011:**
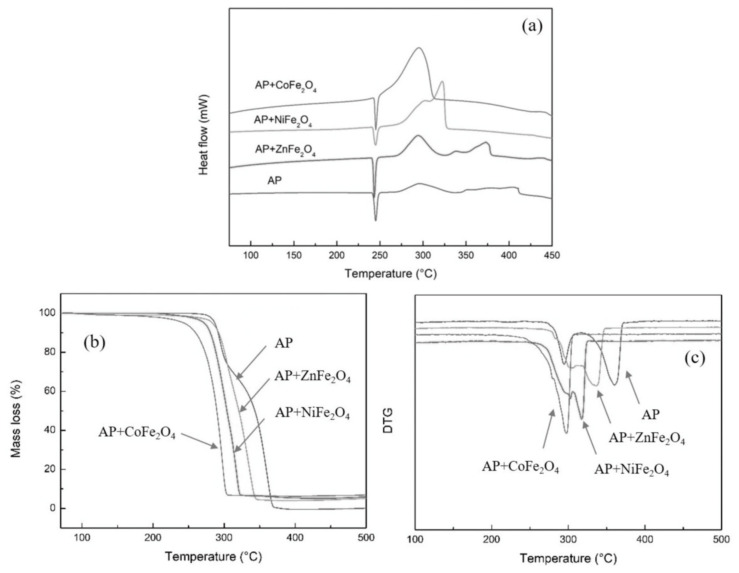
DSC and TG-DTG curves of AP before and after mixed with different ferrates at 10 °C min^−1^ (**a**) DSC curves, (**b**) TG curves, (**c**) DTG curves. (**a**–**c**) Reproduced with the permission of ref. [79]. Copyright 2020, Propellants Explosives Pyrotechnics.

**Figure 12 molecules-28-02035-f012:**
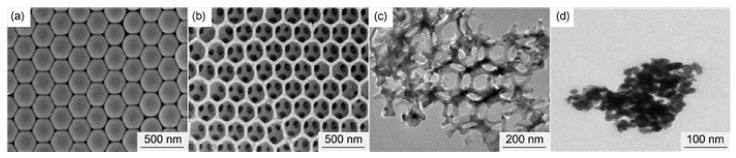
SEM image of (**a**) colloidal crystal template of polystyrene (PS); (**b**) SEM image of nanoporous CoFe_2_O_4_; (**c**) TEM image of nanoporous CoFe_2_O_4_; (**d**) TEM image of nanospheres CoFe_2_O_4_. (**a**–**d**) Reproduced with the permission of ref. [73]. Copyright 2016, Acta Physico-Chimica Sinica.

**Figure 13 molecules-28-02035-f013:**
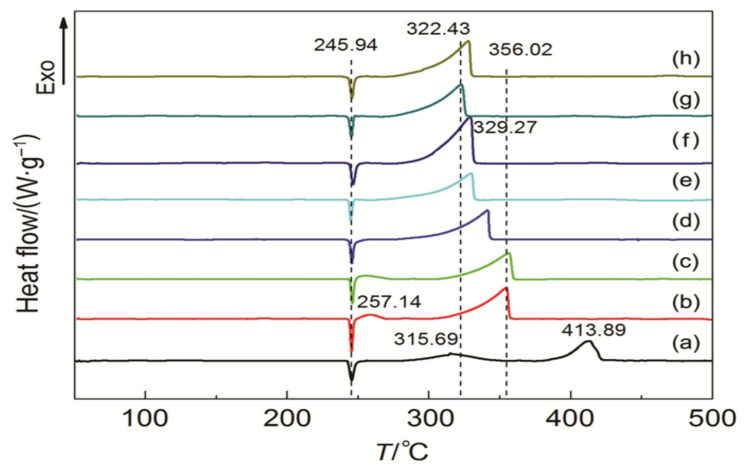
DSC curves of the AP decomposition in the absence and presence of different blend ratios of nanoporous CoFe_2_O_4_. Reproduced with the permission of ref. [73]. Copyright 2016, Acta Physico-Chimica Sinica. (**a**) pure AP; (**b**) AP + nanoporous CoFe_2_O_4_-1% (ω); (**c**) AP + nanoporous CoFe_2_O_4_-2%; (**d**) AP + nanoporous CoFe_2_O_4_-3%; (**e**) AP + nanoporous CoFe_2_O_4_-4%; (**f**) AP + nanoporous CoFe_2_O_4_-5%; (**g**) AP + nanoporous CoFe_2_O_4_-6%; (**h**) AP + nanoporous CoFe_2_O_4_-7%.

**Figure 14 molecules-28-02035-f014:**
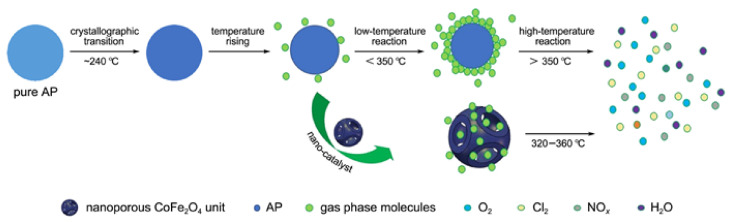
Schematic diagram of the thermal AP decomposition process in the absence and presence of nanoporous CoFe_2_O_4_. Reproduced with the permission of ref. [73]. Copyright 2016, Acta Physico-Chimica Sinica.

**Figure 15 molecules-28-02035-f015:**
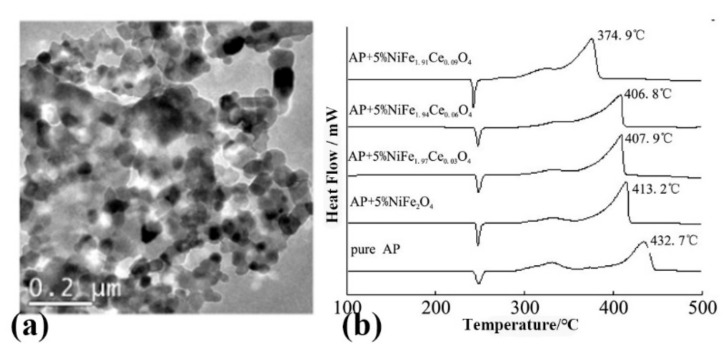
TEM images of the samples (**a**); DSC curves of different samples (**b**). Reproduced with the permission of ref. [81]. Copyright 2012, Journal of Solid Rocket Technology.

**Figure 16 molecules-28-02035-f016:**
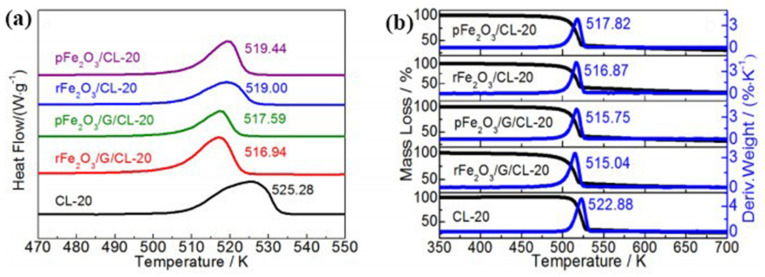
(**a**) DSC and (**b**) TG-DTG curves of CL-20 mixed with different catalysts. (**a**,**b**) Reproduced with the permission of ref. [98]. Copyright 2020, Acta Physico-Chimica Sinica.

**Figure 17 molecules-28-02035-f017:**
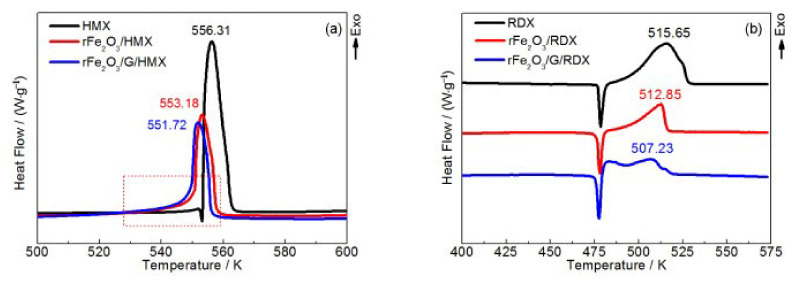
DSC curves of (**a**) rFe_2_O_3_/G and (**b**) rFe_2_O_3_ mixed with HMX and RDX. (**a**,**b**) Reproduced with the permission of ref. [98]. Copyright 2020, Acta Physico-Chimica Sinica.

**Figure 18 molecules-28-02035-f018:**
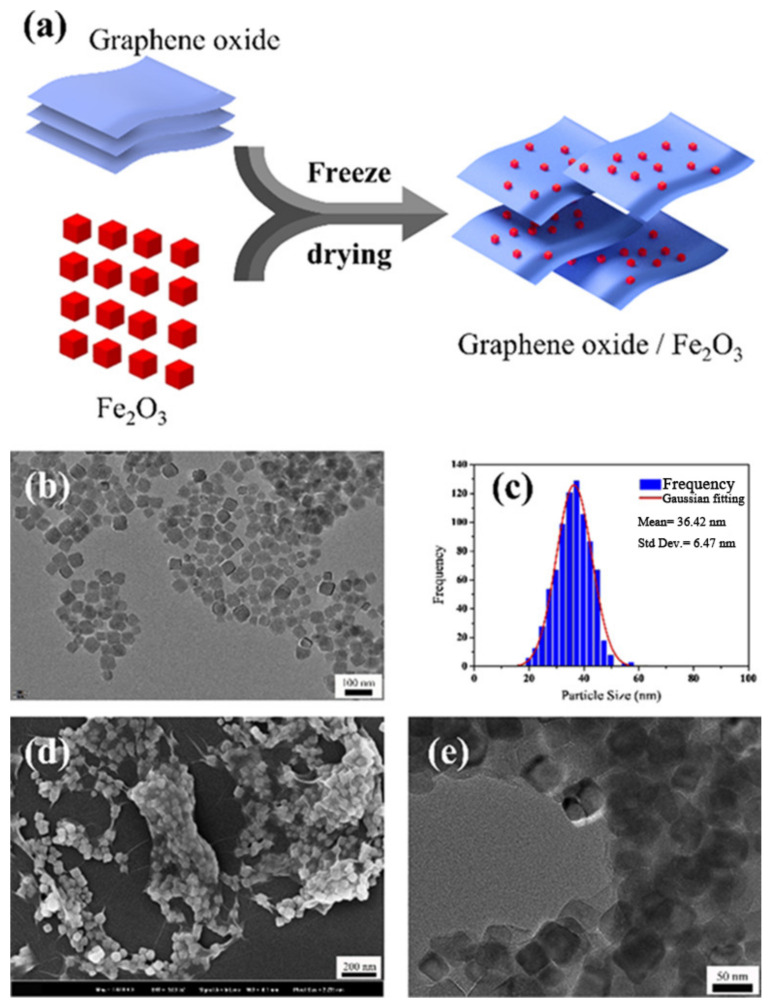
(**a**) Schematic diagram of the preparation and structure of the GO /Fe_2_O_3_ hybrid. (**b**) HRTEM image of the Fe_2_O_3_nanoparticles, (**c**) size distribution of Fe_2_O_3_ nanoparticles. (**d**) SEM image of sample 1, (**e**) HRTEM image of sample 1. (**a**–**c**) Reproduced with the permission of ref. [99]. Copyright 2021, Langmuir.

**Figure 19 molecules-28-02035-f019:**
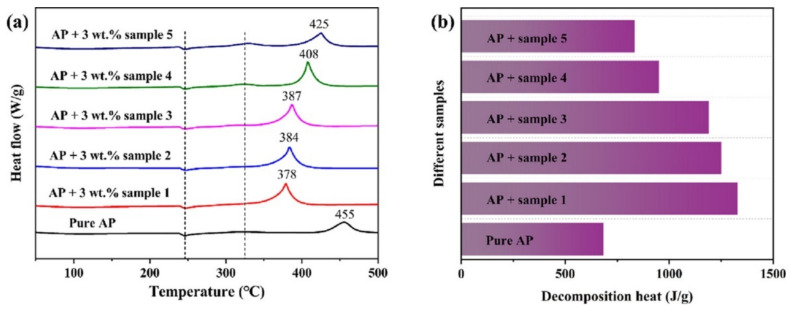
(**a**) DSC curves of AP thermal decomposition with and without the GO/Fe_2_O_3_ nanocomposite; (**b**) heat release during the decomposition of AP with and without catalysts. (**a**,**b**) Reproduced with the permission of ref. [99]. Copyright 2021, Langmuir. Sample 1 GO:Fe_2_O_3_ = 1:10, Sample 2 GO:Fe_2_O_3_ = 3:20, Sample 3 GO:Fe_2_O_3_ = 3:10, Sample 4 GO:Fe_2_O_3_ = 1:1, Sample 5 GO:Fe_2_O_3_ = 3:1.

**Figure 20 molecules-28-02035-f020:**
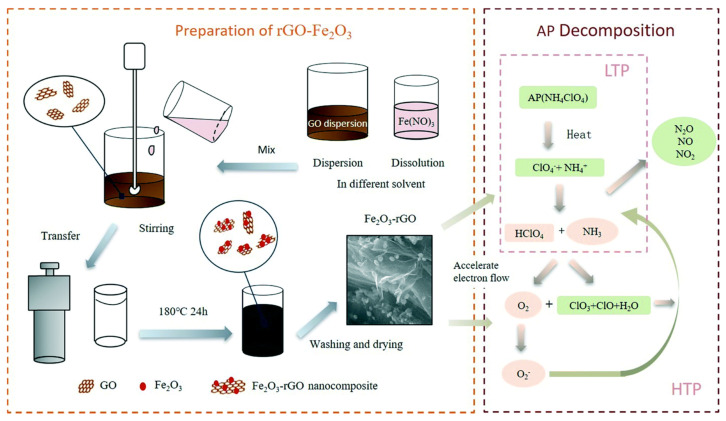
Illustration of Fe_2_O_3_/rGO fabrication and its catalytic mechanism for the thermal decomposition of AP. Reproduced with the permission of ref. [101]. Copyright 2018, Crystengcomm.

**Figure 21 molecules-28-02035-f021:**
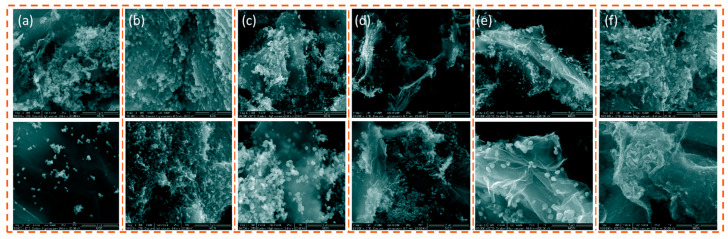
SEM and TEM images of Fe_2_O_3_/rGO composites fabricated in different solvents: (**a**) NBA, (**b**) H_2_O, (**c**) EA, (**d**) NMP, (**e**) DMF, and (**f**) EG. (**a**–**f**) Reproduced with the permission of ref. [101]. Copyright 2018, Crystengcomm.

**Figure 22 molecules-28-02035-f022:**
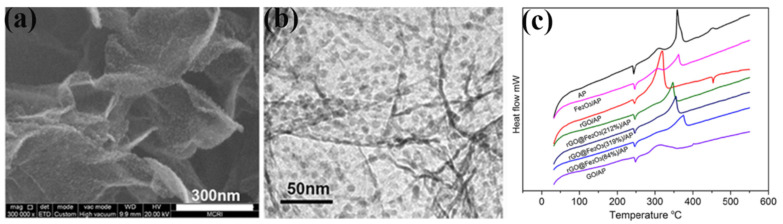
SEM images of (**a**) rGO@(84wt%)Fe_2_O_3_, and TEM images of (**b**) rGO@(84wt%)Fe_2_O_3_; (**c**) DSC curves for the thermal decomposition of AP with different additives. (**a**–**c**) Reproduced with the permission of ref. [104]. Copyright 2018, Applied Surface Science.

**Figure 23 molecules-28-02035-f023:**
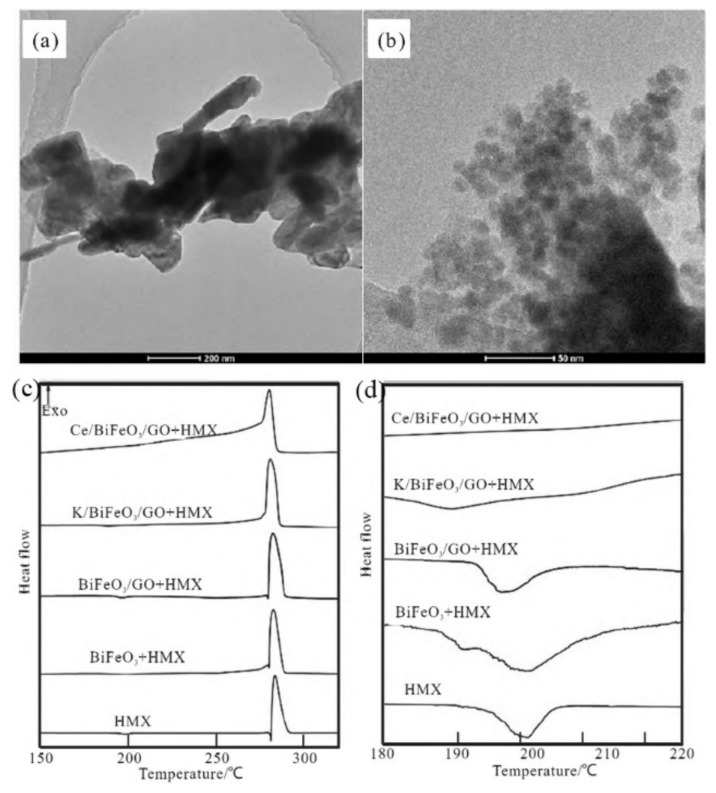
TEM images of (**a**,**b**) K doped BiFeO_3_/GO and Ce doped BiFeO_3_/GO; DSC curves of (**c**,**d**) HMX and the mixtures of HMX with four different BiFeO_3_ composites. (**a**–**d**) Reproduced with the permission of ref. [107]. Copyright 2022, Journal of Solid Rocket Technology.

**Figure 24 molecules-28-02035-f024:**
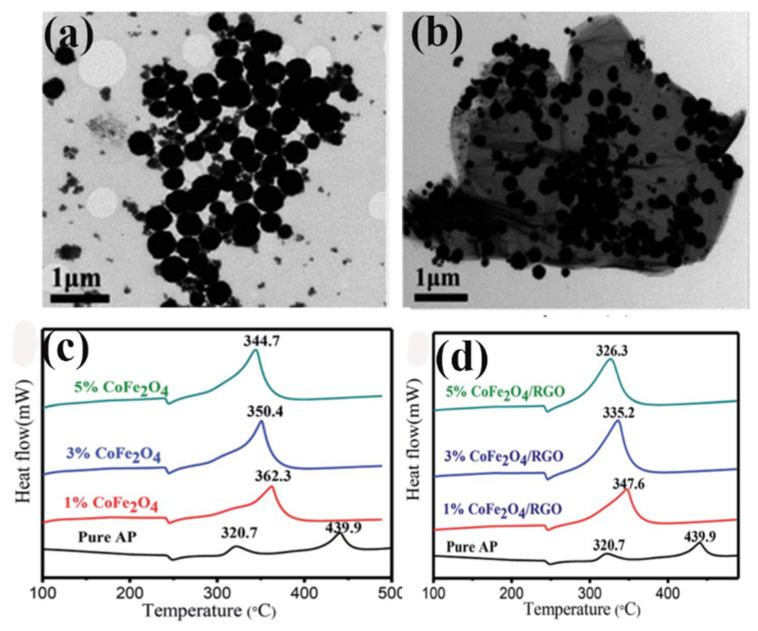
TEM images of (**a**) the as-synthesized CoFe_2_O_4_; (**b**) CoFe_2_O_4_/RGO hybrids. (**c**) DTA curves of pure AP and AP mixed with CoFe_2_O_4_ (1%, 3%, and 5%), (**d**) pure AP and AP mixed with CoFe_2_O_4_/RGO hybrids (1%, 3%, and 5%). (**a**–**d**) Reproduced with the permission of ref. [108]. Copyright 2016, RSC Advances.

**Figure 25 molecules-28-02035-f025:**
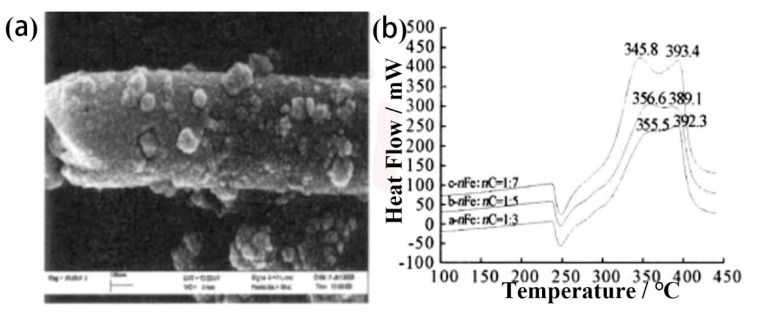
(**a**) SEM image of Fe_2_O_3_/CNTs composite particles; (**b**) DSC curves of AP with different ratio of nFe_2_O_3_: nCNTs. (**a**,**b**) Reproduced with the permission of ref. [114]. Copyright 2008, Journal of Solid Rocket Technology. Cui et al. [115] used ferric nitrate as raw material, citric acid as chelating agent, adjusted the pH value with ammonia water to obtain a uniformly dispersed CNTs sol, and prepared nano-Fe_2_O_3_/CNTs catalyst by the sol-gel method. The nanoscaled Fe_2_O_3_ particles are found to be uniformly coated on the surface of CNTs. The effect of Fe_2_O_3_/CNTs on the thermal decomposition performance of RDX was investigated by DSC. The 5 wt% added Fe_2_O_3_/CNTs composite particles were capable of reducing the decomposition peak temperature and the activation energy of RDX by 14.1 °C and 42.03kJ·mol^−1^, respectively.

**Figure 26 molecules-28-02035-f026:**
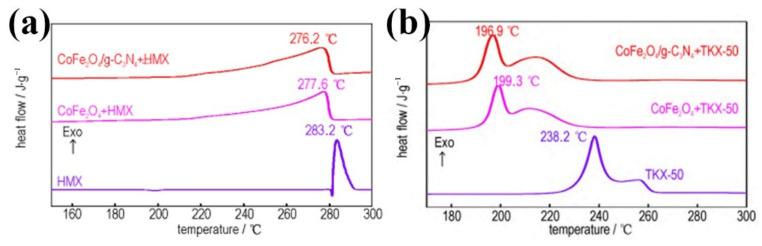
DSC curves of (**a**) HMX, CoFe_2_O_4_+HMX and CoFe_2_O_4_/g-C_3_N_4_+HMX; (**b**) DSC curves of TKX-50, CoFe_2_O_4_+TKX-50 and CoFe_2_O_4_/g-C_3_N_4_+TKX-50. (**a**,**b**) Reproduced with the permission of ref. [122]. Copyright 2022, Journal of Energetic Materials.

**Figure 27 molecules-28-02035-f027:**
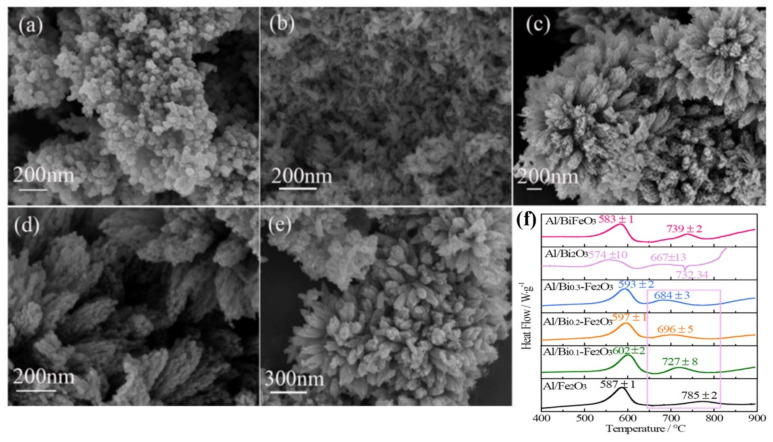
SEM images of (**a**) Fe_2_O_3_ and (**b**) Bi_0_._1_-Fe_2_O_3_; (**c**,**d**) low and high magnification images of Bi_0_._2_-Fe_2_O_3_; (**e**) SEM image of Bi_0_._3_-Fe_2_O_3_; (**f**) DSC curve of six thermites obtained with a heating rate of 10 °C·min^−1^ under N_2_ flow. (**a**–**f**) Reproduced with the permission of ref. [134]. Copyright 2021, Chemical Engineering Journal.

**Figure 28 molecules-28-02035-f028:**
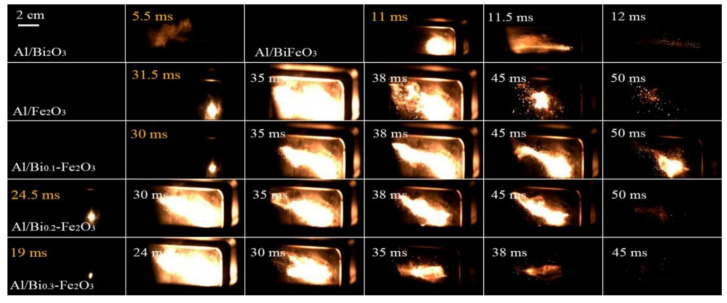
Sequential images of Al/Fe_2_O_3_, Al/Bi doped Fe_2_O_3_, Al/Bi_2_O_3_ and Al/BiFeO_3_ combustion process in ambient condition. Reproduced with the permission of ref. [134]. Copyright 2021, Chemical Engineering Journal.

**Figure 29 molecules-28-02035-f029:**
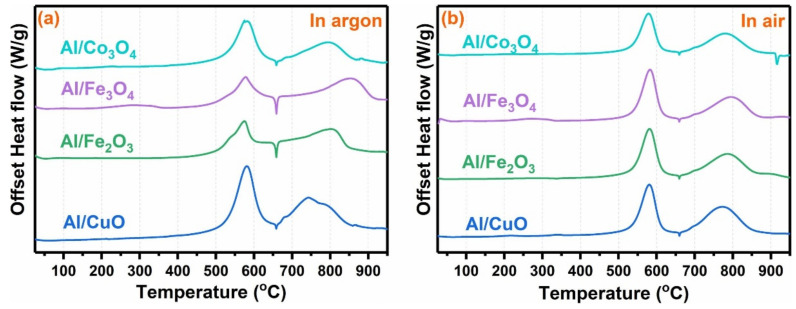
DSC traces collected for Al/CuO, Al/Fe_2_O_3_, Al/Fe_3_O_4_ and Al/Co_3_O_4_ as prepared and ramped in flowing Ar (**a**) and air (**b**) at 10 °C/min. The y-axis values are offset. (**a**,**b**) Reproduced with the permission of ref. [145]. Copyright 2021, Chemical Engineering Journal.

**Figure 30 molecules-28-02035-f030:**
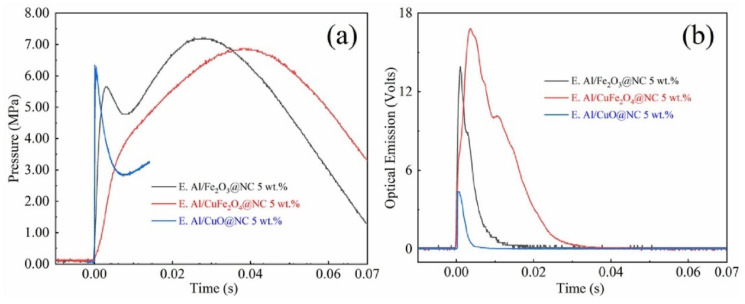
The representative time resolved pressure and optical emission curves of Al/CuO@NC, Al/Fe_2_O_3_@NC and Al/CuFe_2_O_4_@NC composites. (**a**,**b**) Reproduced with the permission of ref. [149]. Copyright 2022, Fuel.

**Figure 31 molecules-28-02035-f031:**
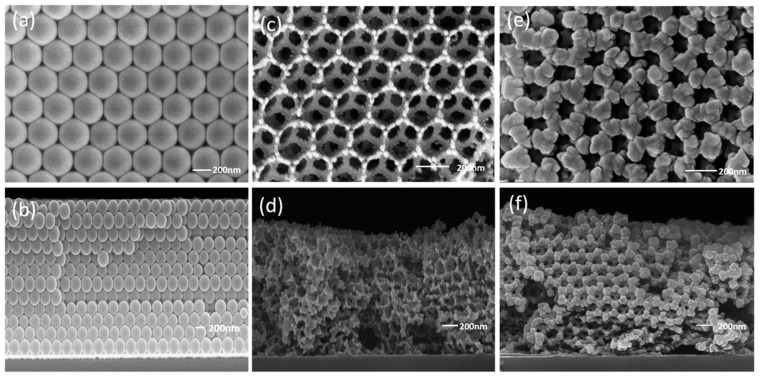
SEM images of (**a**,**b**) PS template, (**c**,**d**) 3DOM NiFe_2_O_4_ membrane and (**e**,**f**) 3DOM Al/NiFe_2_O_4_ membrane after Al deposition, (**a**,**c**,**e**) surface view and (**b**,**d**,**f**) cross-section view. (**a**–**f**) Reproduced with the permission of ref. [150]. Copyright 2016, RSC Advances.

**Figure 32 molecules-28-02035-f032:**
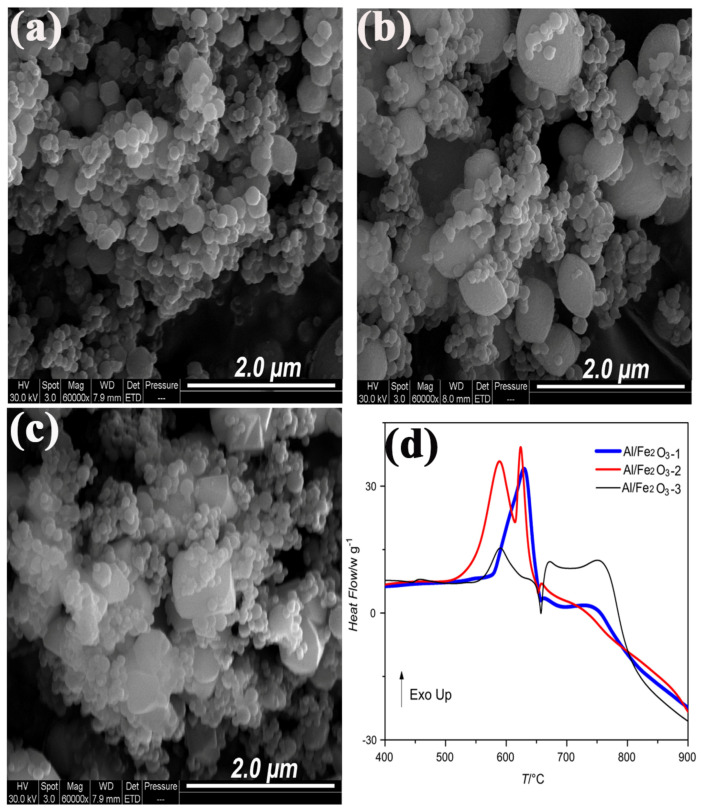
SEM images of (**a**) Al/Fe_2_O_3_-granular, (**b**) Al/Fe_2_O_3_-oval, (**c**) Al/Fe_2_O_3_-polyhedral, (**d**) DSC curves of the three thermites obtained at a heating rate of 10 °C·min^−1^ (1-granular, 2-oval, 3-polyhedral). (**a**–**d**) Reproduced with the permission of ref. [8]. Copyright 2014, Journal of Solid State Chemistry.

**Figure 33 molecules-28-02035-f033:**
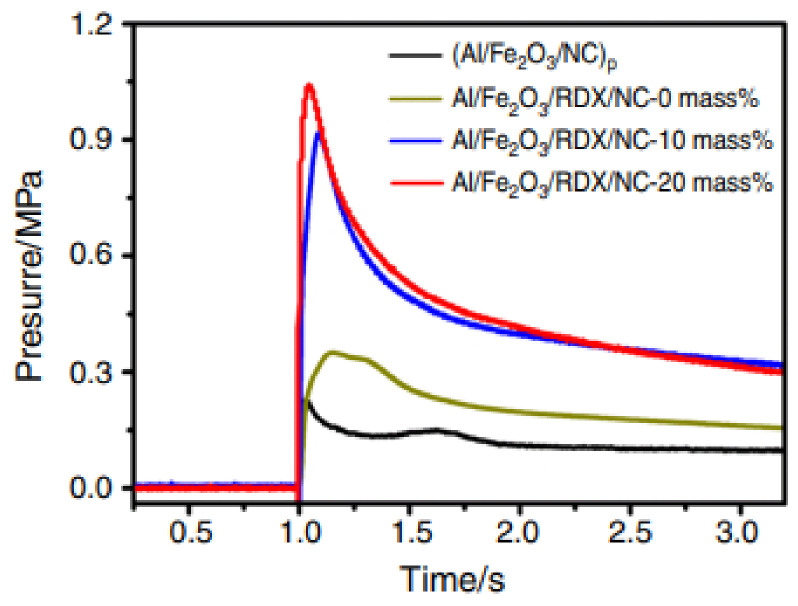
Temporal pressure rise curves during the reaction of (Al/Fe_2_O_3_/NC)_p_ composites prepared by simple physical mixing and Al/Fe_2_O_3_/RDX/NC composites containing different RDX contents prepared by electrospray process. Reproduced with the permission of ref. [151]. Copyright 2019, Journal of Thermal Analysis and Calorimetry.

**Figure 34 molecules-28-02035-f034:**
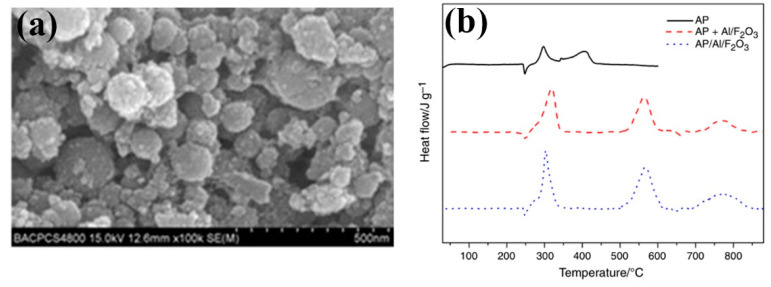
(**a**) SEM images of Al/Fe_2_O_3_; (**b**) DSC curves of AP, AP + Al/Fe_2_O_3_, AP/Al/Fe_2_O_3_ at 10 °C·min^−1^. (**a**,**b**) Reproduced with the permission of ref. [152]. Copyright 2014, Journal of Thermal Analysis and Calorimetry.

**Figure 35 molecules-28-02035-f035:**
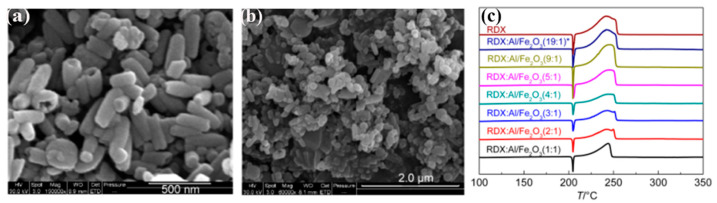
(**a**,**b**) SEM images of nano-Fe_2_O_3_ and super thermite Al/Fe_2_O_3_; (**c**) DSC curves of RDX and RDX/super thermite (Al/Fe_2_O_3_) mixture. (**a**–**c**) Reproduced with the permission of ref. [153]. Copyright 2013, Acta Physico-Chimica Sinica.

**Figure 36 molecules-28-02035-f036:**
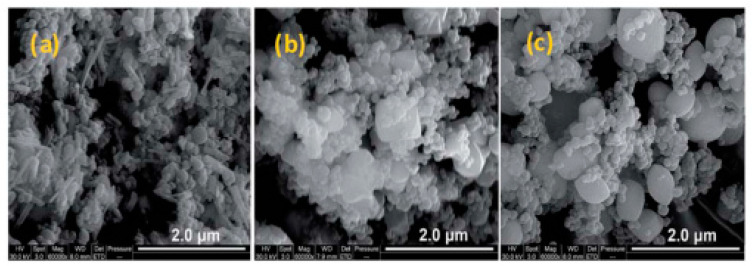
SEM images of Al/Fe_2_O_3_. (**a**) Al/Fe_2_O_3_ (r); (**b**) Al/Fe_2_O_3_ (p) and (**c**) Al/Fe_2_O_3_ (o). (**a**–**c**) Reproduced with the permission of ref. [154]. Copyright 2017, RSC Advances.

**Figure 37 molecules-28-02035-f037:**
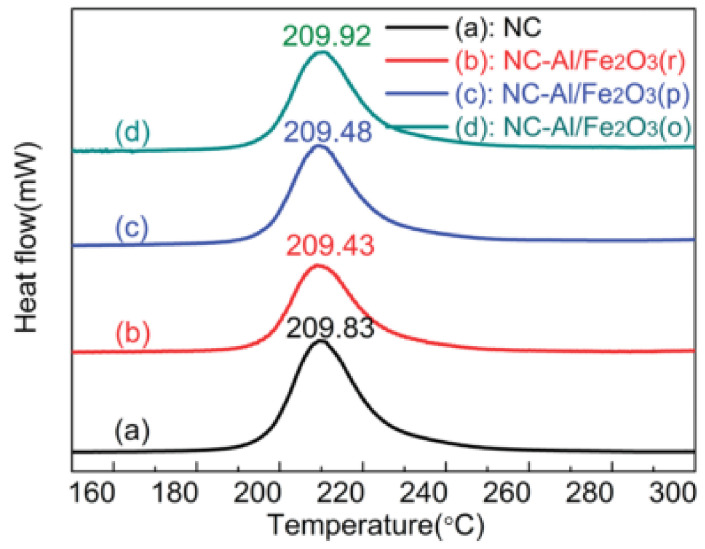
DSC curves of NC-Al/Fe_2_O_3_ and NC. Reproduced with the permission of ref. [154]. Copyright 2017, RSC Advances.

**Figure 38 molecules-28-02035-f038:**
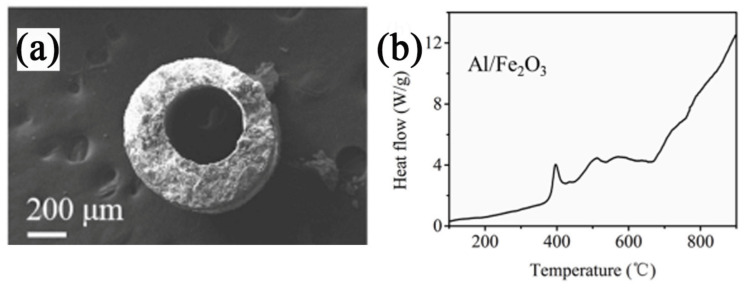
(**a**) SEM images of Al/Fe_2_O_3_/F2311; (**b**) DSC curves of Al/Fe_2_O_3_/F2311. (**a**,**b**) Reproduced with the permission of ref. [155]. Copyright 2021, Materials Chemistry and Physics.

**Table 1 molecules-28-02035-t001:** Summary of thermal behavior parameters of EMs with different morphologies of hematite.

Morphologies	Samples	Preparation Method	Mass Ratios	*T* _p_	Δ*H*	*E*	Refs.
	AP	assistance of F^−^	0	435.0	-	-	[35]
dodecahedral	Fe_2_O_3_/AP		2%	415.0	-	-	
	AP	one-pot	0	436.4	-	-	[36]
nanorods	Fe_2_O_3_/AP		1%	387.5	-	-	
micro-octahedrons	Fe_2_O_3_/AP		1%	426.4	-	-	
	CL-20	hydrothermal	0	248.8	-	181.1	[38]
tetrakaidecahedral	Fe_2_O_3_/CL-20		33.33%	243.3	-	177.8	
grainy	Fe_2_O_3_/CL-20		33.33%	243.9	-	176.1	
	NC	hydrothermal	0	208.6	-	(17 ± 2) x 10 + 01	[39]
granular	Fe_2_O_3_/NC/Cl		5%	209.2	-	(16 ± 2) x 10 + 01	
	Fe_2_O_3_/NC/N		5%	209.4	-	(17 ± 3) x 10 + 01	
	Fe_2_O_3_/NC/S		5%	209.1	-	(17 ± 1) x 10 + 01	
	NC	hydrothermal	0	209.6	-	-	[40]
granular	Fe_2_O_3_/NC		12.6%	209.9	-	189.9	
	HMX	co-precipitation	0	285.0	-	-	[41]
granular	Fe_2_O_3_/HMX		1%	272.0	-	-	
	AP	hydrothermal	0	428.5	-	-	[42]
nano-rods	Fe_2_O_3_/AP		2%	351.7	-	-	
	AP	hydrothermal	0	434.9	-	245.8 ± 32.4	[43]
rod-like	ro-Fe_2_O_3_/AP		2%	358.1	1517.4	133.6 ± 6.3	
rhombic	rh-Fe_2_O_3_/AP		2%	381.5	1494.8	153.0 ± 8.7	
pseudo-cubic	pc-Fe_2_O_3_/AP		2%	395.0	1469.1	161.2 ± 0.5	
	AP	hydrothermal	0	516.6	-	-	[44]
nano-disks	Fe_2_O_3_/AP		1%	394.0	-	-	
irregular particles	Fe_2_O_3_/AP		1%	419.8	-	-	
	AP	indica (neem) leaf extract.	0	445.0	-	160.0 ± 2.0	[45]
hexagonal cone	Fe_2_O_3_/AP		1%	370.0	-	112.0 ± 5.0	
	AP	high-gravity reactive precipitation	0	437.4	860.0	218.0	[46]
rhombohedron	G30 Fe_2_O_3_/AP		2%	396.7	984.0	167.1	
	G220 Fe_2_O_3_/AP		2%	384.0	1235.0	163.3	

Note: *T* (°C); Δ*H* (J·g^−1^); *E* (KJ·mol^−1^).

**Table 3 molecules-28-02035-t003:** Summary of thermal behavior parameters of carbon composites EMs.

Materials	Samples	Preparation Method	Mass Ratios	*T* _p_	Δ*H*	*E*	Refs.
GO	Fe_2_O_3_/GO/HMX	hydrothermal	1%	242.6	-	139.9	[97]
G	CL-20	hydrothermal	0	252.1	179.2	183.4	[98]
	pFe_2_O_3_/CL-20		20%	246.3	-	172.6	
	rFe_2_O_3_/CL-20		20%	245.9	161.5	165.6	
	pFe_2_O_3_/G/CL-20		20%	244.4	-	159.9	
	rFe_2_O_3_/G/CL-20		20%	243.8	148.9	153.0	
	HMX		0	283.2	-	-	
	rFe_2_O_3_/HMX		20%	280.0	-	-	
	rFe_2_O_3_/G/HMX		20%	278.6	-	-	
	RDX		0	242.5	-	-	
	rFe_2_O_3_/RDX		20%	239.7	-	-	
	rFe_2_O_3_/G/RDX		20%	234.1	-	-	
GO	AP	vacuum-freeze-drying	0	455.0	-	-	[99]
Sample 1	10:1 Fe_2_O_3_/GO/AP		3%	378.0	-	-	
Sample 2	20:3 Fe_2_O_3_/GO/AP		3%	384.0	-	-	
Sample 3	10:3 Fe_2_O_3_/GO/AP		3%	387.0	-	-	
Sample 4	1:1 Fe_2_O_3_/GO/AP		3%	408.0	-	-	
Sample 5	3:1 Fe_2_O_3_/GO/AP		3%	425.0	-	-	
GO	AP	hydrothermal	0	432.0	-	129.0	[100]
	GO/AP		1.84%	400.0	-	100.1	
	Fe_2_O_3_/AP		1.84%	397.0	-	89.3	
	Fe_2_O_3_/G/AP		1.84%	380.0	-	80.3	
GO	AP	solvothermal	0	440.8	-	290.0	[101]
	GO/AP		5%	372.5	-	216.2	
H_2_O	rGO/AP		5%	360.8	-	128.8	
H_2_O	Fe_2_O_3_/AP		5%	369.4	-	170.3	
DMF	Fe_2_O_3_/rGO/AP		5%	321.2	-	119.8	
rGO	AP	atomic layer deposition	0	440 ± 5	-	-	[104]
	Fe_2_O_3_/rGO/AP		5%	358	-	-	
rGO	AP	direct precipitation	0	425.8	-	-	[105]
	Fe_2_O_3_/rGO/AP		1%	413.1	-	-	
rGO	AP	hydrothermal	0	462.0	887.0	172.1	[106]
	BiFeO_3_/AP		-	449.0	-	144.5	
	1% BiFeO_3_/rGO/AP		-	427.0	-	-	
	2% BiFeO_3_/rGO/AP		-	396.0	-	-	
	3% BiFeO_3_/rGO/AP		-	354.0	-	-	
	4% BiFeO_3_/rGO/AP		-	295.0	2518.0	128.4	
	5% BiFeO_3_/rGO/AP		-	308.0	-	-	
GO	HMX	hydrothermal	0	283.4	-	-	[107]
	BiFeO_3_/HMX		20%	282.7	-	-	
	BiFeO_3_/GO/HMX		20%	282.4	-	-	
	K/BiFeO_3_/GO/HMX		20%	280.9	-	-	
	Ce/BiFeO_3_/GO/HMX		20%	280.5	-	-	
rGO	AP	one-pot solvothermal	0	439.9	-	147.6	[108]
	1% CoFe_2_O_4_/AP		-	362.3	-	-	
	3% CoFe_2_O_4_/AP		-	350.4	-	131.5	
	5% CoFe_2_O_4_/AP		-	344.7	-	-	
	1% CoFe_2_O_4_/rGO/AP		-	347.6	-	-	
	3% CoFe_2_O_4_/rGO/AP		-	335.2	-	117.9	
	5% CoFe_2_O_4_/rGO/AP		-	326.3	-	-	
rGO	AP	solvothermal	0	439.9	-	147.6	[109]
	3% rGO/AP		-	433.7	-	-	
	3% NiFe_2_O_4_/AP		-	363.4	-	-	
	1% NiFe_2_O_4_/rGO/AP		-	359.2	-	-	
	3% NiFe_2_O_4_/rGO/AP		-	347.3	-	128.3	
	5% NiFe_2_O_4_/rGO/AP		-	336.1	-	-	
rGO	AP	one-step hydrothermal	0	424.7	-	160.3	[110]
	ZnFe_2_O_4_/AP		1%	380.7	-	-	
	ZnFe_2_O_4_/AP		3%	365.0	-	-	
	ZnFe_2_O_4_/AP		5%	360.8	-	143.7	
	ZnFe_2_O_4_/rGO/AP		1%	380	-	-	
	ZnFe_2_O_4_/rGO/AP		3%	367.7	-	-	
	ZnFe_2_O_4_/rGO/AP		5%	354.6	-	127.6	
CNTs	AP	sol-gel	0	478.1	-	178.0	[114]
	simple mix Fe_2_O_3_/CNTs/AP		2%	390.9	-	134.0	
	particle composite Fe_2_O_3_/CNTs/AP		2%	376.0	-	118.0	
CNTs	RDX	sol-gel	0	240.9	-	109.2	[115]
	Fe_2_O_3_/CNTs/RDX		5%	226.8	-	67.17	
g-C_3_N_4_	AP	direct copolymerization	0	454.4	-	216.0	[120]
	g-C_3_N_4_/AP		10%	384.4	-	119.8	
g-C_3_N_4_	AP	In situ	0	437.4	-	-	[121]
	1% Fe_2_O_3_/C_3_N_4_/AP		2%	376.2	-	-	
	2% Fe_2_O_3_/C_3_N_4_/AP		2%	368.5	-	-	
	3% Fe_2_O_3_/C_3_N_4_/AP		2%	358.9	-	-	
	5% Fe_2_O_3_/C_3_N_4_/AP		2%	348.1	-	-	
g-C_3_N_4_	HMX	solvothermal	0	283.2	-	502.2	[122]
	CoFe_2_O_4_/HMX		20%	277.6	-	225.6	
	CoFe_2_O_4_/g-C_3_N_4_/HMX			276.2	-	161.1	
	TKX-50			238.2	-	172.1	
	CoFe_2_O_4_/TKX-50			199.3	-	159.8	
	CoFe_2_O_4_/g-C_3_N_4_/TKX-50			196.9	-	151.1	

Note: *T* (°C); Δ*H* (J·g^−1^); *E* (kJ·mol^−1^).

**Table 4 molecules-28-02035-t004:** Parameters of some nano-thermites.

Materials	Preparation Method	Application	*T* _exo1_	*T* _exo2_	Δ*H*	Ref.
Al/B/Fe_2_O_3_	sol-gel synthetic	thermite	582.0	790.0	3270.0	[133]
Al/B/Fe_2_O_3_	physical blending		559.0	780.0	2610.0	
Al/Fe_2_O_3_	one-step hydrothermal	solid propellants	587.0 ± 1.0	785.0 ± 2.0	3125.0 ± 31.0	[134]
Al/Bi_0_._1_-Fe_2_O_3_			602.0 ± 2.0	727.0 ± 8.0	2893.0 ± 16.0	
Al/Bi_0_._2_-Fe_2_O_3_			597.0 ± 1.0	696.0 ± 5.0	2720.0 ± 22.0	
Al/Bi_0_._3_-Fe_2_O_3_			593.0 ± 2.0	684.0 ± 3.0	2349.0 ± 14.0	
Al/Bi_2_O_3_			574.0 ± 10.0	667.0 ± 13.0	852.0 ± 34.0	
Al/BiFeO_3_			583.0 ± 1.0	739.0 ± 2.0	1812.0 ± 23.0	
in Ar Al/CuO	virtual thermal aging	thermite	528.0	691.0	3945.0	
in Ar Al/Fe_2_O_3_			514.0	733.0	2791.0	[145]
in Ar Al/Fe_3_O_4_			527.0	770.0	2516.0	
in Ar Al/Co_3_O_4_			534.0	698.0	2457.0	
in air Al/CuO			538.0	697.0	5125.0	
in air Al/Fe_2_O_3_			539.0	705.0	4214.0	
in air Al/Fe_3_O_4_			540.0	710.0	4302.0	
in air Al/Co_3_O_4_			536.0	702.0	3958.0	
Al/CuFe_2_O_4_@NC	electrospray		567.8	-	2064.6	[149]
Al/Fe_2_O_3_@NC 5wt%			576.5	-	2348.5	
Al/Cuo@NC 5wt%			598.8	-	1763.9	
Al/NiFe_2_O_4_	templating	MEMS pyrotechnics	359.2	594.6	2921.7	[150]
Al/Fe_2_O_3_/NC	electrospray	thermite	201.4	587.8	-	[151]
Al/Fe_2_O_3_/RDX/NC-0			206.2	589.6	-	
Al/Fe_2_O_3_/RDX/NC-10			224.7	581.8	-	
Al/Fe_2_O_3_/RDX/NC-20			227.7	584.6	-	
Al/Fe_2_O_3_+AP	physicalmixing	micro-propellants	-	-	1820.0	[152]
Al/Fe_2_O_3_/AP	solvent-anti-solvent		-	-	1400.0	
Al/Fe_2_O_3_/RDX (1:19)	hydrothermal	super thermites	241.6	-	-	[153]
Al/Fe_2_O_3_/RDX (1:9)			246.3	-	-	
Al/Fe_2_O_3_/RDX (1:5)			246.0	-	-	
Al/Fe_2_O_3_/RDX (1:4)			245.9	-	-	
Al/Fe_2_O_3_/RDX (1:3)			242.5	250.9	-	
Al/Fe_2_O_3_/RDX (1:2)			239.6	247.2	-	
Al/Fe_2_O_3_/RDX (1:1)			244.5	-	-	
Al/Fe_2_O_3_(p)/NC	hydrothermal	micro-propellants	209.5	-	-	[154]
Al/Fe_2_O_3_(o)/NC			209.9	-	-	
Al/Fe_2_O_3_(r)/NC			209.4	-	-	
Al/CuO/F2311	3D printing		288.0	-	1500.0	
Al/Fe_2_O_3_/F2311			368.0	-	1552.0	
Al/Bi_2_O_3_/F2311			410.0	-	1071.0	
Al/PTFE/F2311			377.0	-	7200.0	

Note: *T*_exo1_ is the first exothermic temperature(°C); *T*_exo2_ is the second exothermic temperature(°C); Δ*H* is the energy output (J·g^−1^).

## Data Availability

Not applicable.

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
