# Peer review of "Hematite: A Good Catalyst for the Thermal Decomposition of Energetic Materials and the Application in Nano-Thermite"

_molecules, 2023, doi:10.3390/molecules28052035_

Round 1
Reviewer 1 Report
This timely review summarizes the reported research works on the synthesis of heamatite through various routes and their well known catalytic activity on the decomposition of several energetic materials such as AP, RDX, HMX, and CL-20. The catalytic activity has been extensively discussed as a function of the morphology of the heamatite obtained through different synthetic routes. The review also delves into the relatively new area of the haematite composites of carbon nanomaterials such as graphene and carbon nitride and their prospective application as catalyst systems for thermal decomposition of AP. The utility of nanothermites to the field of energetic materials is also discussed in the final section. The authors should be credited for extensively covering the literature in all these areas and presenting the review with sufficient clarity. I have the following specific comments on the review for further improvements.
1. The scope of the review should be clearly indicated in the introduction section.
2. The application of heamatite as a ballistic modifier for propellants is well studied in the context as a viable alternative to the widely used lead based compounds. The authors should consider including a section on the application of haematite as a burn rate modifier in propellant formulations.
3. The section 5 on nanothermites and their application appears to be an outlier from the general theme covered under the review. This discussion should be made more confined to the use of heamatite of different morphologies in nanothermite formulations and related properties. The title of the section should also be appropriately changed .
4. It is suggested that the conclusion section be replaced by a future outlook section to reflect upon the potential futuristic application of haematite materials to energetic materials and technology
5. The English and grammar are largely OK. Certain statements need to be rephrased for the sake of clarity : a) Page 2; Lines 120-22, b)Page 5; Lines 189-90,
6) Page 7, Line 265; Briefly explain what is G
7) Page 9, Line 294, word "particularity" should be replaced by specificity.
Reviewer 2 Report
Please see the attachment.

Reviewer 3 Report
The manuscript discusses a number of issues of the catalytic effect of iron oxide (hematite) and several other compounds on the decomposition of energetic materials, as well as the use of hematite in nanothermites. The main results of the research were obtained on the basis of calorimetric and thermogravimetric measurements of thermal decomposition during slow heating. A large array of experimental data has been considered.
However, a number of important questions remained outside the scope of consideration. For example, the particle sizes of the catalyst are indicated, but the particle size of the energetic material is not indicated, although it is obvious that the effect of catalysts may be different for large and small particles. There are also no even approximate estimates of the influence of the density and mass of the samples under study. Is it possible to use the data obtained from the thermal decomposition of loose packed samples weighing several milligrams to predict the combustion and explosive properties of real energetic compositions?
Additional comments on the manuscript
1. Figure 2-1.
You should edit the caption to the figure. The designations (a), (b) and (c) are used both for designating images and for DSC curves, twice in different parts of the figure. The DSC curve for pure AP appears to be (a) in Figure (c), but it is not specified
2. Lines 143, 145.
The phrase "The tetrakaidecahedral nano-Fe2O3 did not change the thermal decomposition mechanism of CL-20, while, the grainy nano-Fe2O3 changed the thermal decomposition mechanism of CL-20" is not entirely clear. Additives of both types reduce the appearance of the exothermic peak by about 5 K. Thus, their effect on the mechanism is approximately the same.
3. Lines 161-162.
“The catalytic activity of the as-prepared nanoparticles for the thermal decomposition of NC was investigated by DSC. All curves show only one exothermic peak, corresponding to the breaking of the O-NO2 bond". I would like to get a justification why this exothermic peak is attributed to the break of this bond? As a rule, the primary reactions during the decomposition of nitro compounds are endothermic in nature, the breaking of chemical bonds requires energy. Exothermic reactions occur in the secondary reactions of the formation of end products CO, H2O, etc.
